# Allosteric gate modulation confers K⁺ coupling in glutamate transporters

Daniel Kortzak[1] (ID), Claudia Alleva[1], Ingo Weyand[1], David Ewers[1,2,3], Meike I Zimmermann[1], Arne Franzen[1], Jan-Philipp Machtens[1,4,†,*] (ID) & Christoph Fahlke[1,†,**] (ID)

## Abstract

Excitatory amino acid transporters (EAATs) mediate glial and neuronal glutamate uptake to terminate synaptic transmission and to ensure low resting glutamate concentrations. Effective glutamate uptake is achieved by cotransport with 3 Na⁺ and 1 H⁺, in exchange with 1 K⁺. The underlying principles of this complex transport stoichiometry remain poorly understood. We use molecular dynamics simulations and electrophysiological experiments to elucidate how mammalian EAATs harness K⁺ gradients, unlike their K⁺-independent prokaryotic homologues. Glutamate transport is achieved via elevator-like translocation of the transport domain. In EAATs, glutamate-free re-translocation is prevented by an external gate remaining open until K⁺ binding closes and locks the gate. Prokaryotic $Glt_{Ph}$ contains the same K⁺-binding site, but the gate can close without K⁺. Our study provides a comprehensive description of K⁺-dependent glutamate transport and reveals a hitherto unknown allosteric coupling mechanism that permits adaptions of the transport stoichiometry without affecting ion or substrate binding.

**Keywords** allosteric coupling; excitatory amino acid transporters; K⁺ binding; secondary active transport; transport stoichiometry

**Subject Categories** Membranes & Trafficking; Structural Biology

**The EMBO Journal (2019) 38: e101468**

## Introduction

Glutamate is the major excitatory neurotransmitter in the mammalian central nervous system. Glutamatergic synaptic transmission is terminated by the rapid uptake of glutamate into glial and neuronal cells by glutamate transporters belonging to the excitatory amino acid transporter (EAAT) family (Vandenberg & Ryan, 2013). EAATs mediate coupled cotransport of one glutamate, three Na⁺, and one H⁺ in exchange with one K⁺, and this complex transport stoichiometry permits adjustment of resting synaptic glutamate concentrations in the low nanomolar range (Zerangue & Kavanaugh, 1996; Vandenberg & Ryan, 2013). Related bacterial and prokaryotic transporters (e.g., $Glt_{Ph}$, $Glt_{Tk}$, or $Glt_{Ec}$; here generically referred to as $Glt_{X}$) use Na⁺ or H⁺ cotransport only, illustrating that obligate K⁺ coupling developed later in evolution (Ryan *et al*, 2009; Groeneveld & Slotboom, 2010). EAATs/$Glt_{X}$ are assembled as trimers, with each subunit independently transporting glutamate via alternating inward- and outward-directed translocation of the transport domain (Yernool *et al*, 2004; Boudker *et al*, 2007; Reyes *et al*, 2009; Verdon *et al*, 2014; Guskov *et al*, 2016). Translocation in EAATs is possible with the transport domain either bound to (i) glutamate, three Na⁺, and one H⁺, or (ii) one K⁺ (Zerangue & Kavanaugh, 1996; Appendix Fig S1), but not in the apo state, thus ensuring obligate K⁺ coupling. In contrast, apo state re-translocation is possible in $Glt_{X}$, permitting K⁺-independent transport (Ryan *et al*, 2009).

K⁺-coupled re-translocation is the rate-limiting step in EAATs (Grewer *et al*, 2000, 2012); in K⁺-independent $Glt_{X}$, substrate-bound translocation is much slower (Akyuz *et al*, 2013; Ruan *et al*, 2017). This difference suggests that glutamate-bound translocation has undergone extensive evolutionary optimization in order to increase transport rates and that only K⁺-dependent re-translocation permitted further improvement of transport activity. Under physiological conditions, the extended EAAT transport stoichiometry ensures continuous and increased glutamate uptake (Grewer *et al*, 2012) and efficient glutamatergic signaling in the brain. However, K⁺ coupling has detrimental effects in pathological conditions with reduced energy supply to the brain such as ischemia, where increased external [K⁺] severely impairs or even reverses glutamate transport (Grewer *et al*, 2008).

Thus far, the molecular basis of K⁺ coupling in EAATs has remained enigmatic. The existence of one Na⁺-dependent and one K⁺-dependent hemicycle predicts a K⁺-selective binding site within the transport domain of EAATs (Kanner & Bendahan, 1982; Pines & Kanner, 1990). Several mutations were reported that affected

1 Institute of Complex Systems, Zelluläre Biophysik (ICS-4) and JARA-HPC, Forschungszentrum Jülich, Jülich, Germany
2 Klinik für klinische Neurophysiologie, Universitätsmedizin Göttingen, Göttingen, Germany
3 Abteilung für Neurogenetik, Max-Planck-Institut für Experimentelle Medizin, Göttingen, Germany
4 Department of Molecular Pharmacology, RWTH Aachen University, Aachen, Germany
   *Corresponding author. Tel: +49 2461 614043; E-mail: j.machtens@fz-juelich.de
   **Corresponding author. Tel: +49 2461 613016; E-mail: c.fahlke@fz-juelich.de
   †These authors contributed equally to this work

$K^+$ coupling in mammalian EAATs (Kavanaugh *et al*, 1997; Bendahan *et al*, 2000; Rosental *et al*, 2006; Teichman *et al*, 2009); the mutated residues are broadly scattered across the protein, and it has remained unclear whether they affect $K^+$ binding or other conformational changes necessary for $K^+$-bound re-translocation. Moreover, many of the residues are conserved across EAAT and $Glt_X$ isoforms and therefore cannot explain why EAATs are $K^+$ coupled unlike $Glt_X$. The recently resolved structure of EAAT1 bound to $Na^+$ and aspartate exhibited an overall protein fold very similar to $Glt_X$ and did not uncover the molecular basis of $K^+$-coupled glutamate transport (Canul-Tec *et al*, 2017). We here combine molecular dynamics (MD) simulations of $Glt_{Ph}$ and human EAAT1 with experiments on $Glt_{Ph}$, EAAT1, and EAAT2 to identify the $K^+$-coupling mechanism in EAATs.

# Results

## Unguided MD simulations identify conserved K⁺-binding sites in EAAT1 and Glt_Ph

We conducted all-atom MD simulations starting from the outward-facing conformations (OFC) of apo $Glt_{Ph}$ (Boudker *et al*, 2007; Verdon *et al*, 2014) and human EAAT1 (Canul-Tec *et al*, 2017), embedded in a lipid bilayer and surrounded by an aqueous KCl solution, to monitor spontaneous $K^+$ association in absence of an applied gradient. Apo states were modeled by removing bound $Na^+$ or aspartate molecules from the structures. Since mammalian and prokaryotic transporters operate over a broad range of internal and external $K^+$ concentrations, we tested [KCl] from < 1 mM up to 1 M (Appendix Table S1). These simulations demonstrated that high ionic concentrations do not affect protein dynamics (Appendix Fig S3A and B) and identified the same $K^+$-interaction sites regardless of [$K^+$] (Appendix Fig S3C).

Thousands of spontaneous $K^+$-binding/unbinding events within a total simulation time of more than 185 μs unequivocally defined three common $K^+$-interaction sites (referred to as K1–K3) and the EAAT1-specific K4 site (Fig 1). Simulations of $Glt_{Ph}$ in the inward-facing conformation (IFC; Reyes *et al*, 2009) revealed that $K^+$ associates with the same K1–K3 sites from the cytoplasmic side. The K1 site is buried deep within the transport domain between the Na1 and Na3 sites (Boudker *et al*, 2007; Guskov *et al*, 2016) and is coordinated by main chain carbonyls of G394, N398 in EAAT1 (both TM7; homologous $Glt_{Ph}$ residues G306 and N310), and N483 (TM8; $Glt_{Ph}$ N410); by the β-carboxylates of D400 (TM7; D312) and D487 (TM8; $Glt_{Ph}$ D405); and by the side chain carbonyls of N398 (TM7; $Glt_{Ph}$ N310). The K2 site is located at the tip of hairpin 2 (HP2) and is formed by the main chain carbonyls of T396 (TM7; $Glt_{Ph}$ T303), and S436, I437, and A439 (HP2; $Glt_{Ph}$ S349, I350, A352). The K3 site is located slightly below the kink of TM8 and is coordinated by the β-carboxylates of D476 and T480 (both TM8; $Glt_{Ph}$ D394 and T398), and the main chain carbonyl of S363 (HP1, $Glt_{Ph}$ S276). The EAAT1 K4 site is located between the HP2 helices and formed by the backbone carbonyls of A436, G447, and T450 (HP2; $Glt_{Ph}$ A348, G359, T362), as well as the hydroxyl group of T450 ($Glt_{Ph}$ T362) and the γ-carboxylate of E406 (TM7; $Glt_{Ph}$ Q318). K2 overlaps with the Na2 site, but K1, K3, and K4 differ from the $Na^+$ sites identified by crystallography (Boudker *et al*,

2007; Guskov *et al*, 2016) with K1 being coordinated by residues of both Na1 and Na3 (Fig 1).

## Transient binding to K2 and K3 accelerates occupation of the K1 site

For all tested [$K^+$] between < 1 mM up to 1 M (Appendix Table S1), we observed frequent binding and unbinding events for the K2–K4 sites that preceded stable binding of $K^+$ ions to K1. We performed extensive simulations at 1 M KCl to increase the number of $K^+$ binding events and to obtain converged statistics. Our simulations sampled 122 K1 binding events for EAAT1 and 53/103 events for $Glt_{Ph}$ (OFC/IFC, respectively), without any unbinding during a total observation time of 70/54 μs for $Glt_{Ph}$ (OFC/IFC), and four unbinding events from K1 in EAAT1 within 83 μs. $K^+$ ions never associated directly to K1 in EAAT1 and rarely in $Glt_{Ph}$ (twice in $Glt_{Ph}$ OFC and three times in $Glt_{Ph}$ IFC), but rather via transient interactions with the K2–K4 sites. For both $Glt_{Ph}$ and EAAT1, $K^+$ ions usually associated to the K2 site first; however, K2 binding was unstable, with subsequent relocation and stable binding to the neighboring K1 site (EAAT1, 63 times; $Glt_{Ph}$, 38/92 times in OFC/IFC; Fig 2A and B; Appendix Fig S4A and B). For EAAT1, we also observed initial $K^+$ association with K3 (five events), and subsequent relocation to K1; in $Glt_{Ph}$ K1 binding via K3–K1 relocation occurred eight times in the OFC and twice in the IFC. In the remaining 54 EAAT1, five $Glt_{Ph}$ OFC, and six $Glt_{Ph}$ IFC K1-binding events, a $K^+$ ion bound to one of the K2–4 site first, followed by binding of a second $K^+$ ion to K1 and subsequent unbinding of the $K^+$ ion at K2–4 (see Appendix Tables S2 and S3 for a full list of all observed binding events). We quantified $K^+$-binding/unbinding events by dwell time analysis to construct a kinetic model including all observed $K^+$ occupation states. Transition path analysis showed that the reaction pathways involved in transient $K^+$ occupation of the K2 and K3 sites had the highest flux rates from the apo to the K1-bound state (Fig 2C–E and Appendix Fig S4).

## K1 is responsible for K⁺ binding to EAATs and to Glt_Ph

To further define the $K^+$-binding site responsible for $K^+$-coupled glutamate transport, we next determined the (i) apparent $K^+$ affinity, (ii) ability to discriminate between $K^+$ and $Na^+$, (iii) and voltage dependence of $K^+$ binding and $K^+$-bound re-translocation for the K1–K4 sites.

We calculated equilibrium dissociation constants ($K_D$) of around 8, 30, 350, and 1,500 mM for the K1, K2, K3, and K4 sites (Fig 3A and B), respectively, via dwell time analysis of the large amount of spontaneous $K^+$-binding/unbinding events (Appendix Fig S4). The selectivity of K1–K4 for $K^+$ over $Na^+$ was quantified by calculating relative binding free energies ($\Delta\Delta G_{K\rightarrow Na}$) via non-equilibrium alchemical transformations (Appendix Fig S5; Gapsys *et al*, 2015). K2 was most selective: $\Delta\Delta G_{K\rightarrow Na}$ values were 19.1 ± 0.2 kJ/mol and 19.7 ± 0.20 kJ/mol for EAAT1 and $Glt_{Ph}$, respectively, similar to the values reported for other $K^+$-selective binding sites (Thompson *et al*, 2009; Yu *et al*, 2011). K1 was moderately selective: $\Delta\Delta G_{K\rightarrow Na}$ values were 12.4 ± 0.2 and 13.2 ± 0.3 kJ/mol for EAAT1 and $Glt_{Ph}$, respectively. The K3 and K4 sites were least selective: $\Delta\Delta G_{K\rightarrow Na}$ values were 9.9 ± 0.5 and 2.9 ± 0.3 kJ/mol for EAAT1 and $Glt_{Ph}$ K3 sites and 2.3 ± 0.4 for the EAAT1 K4 site (Fig 3C).

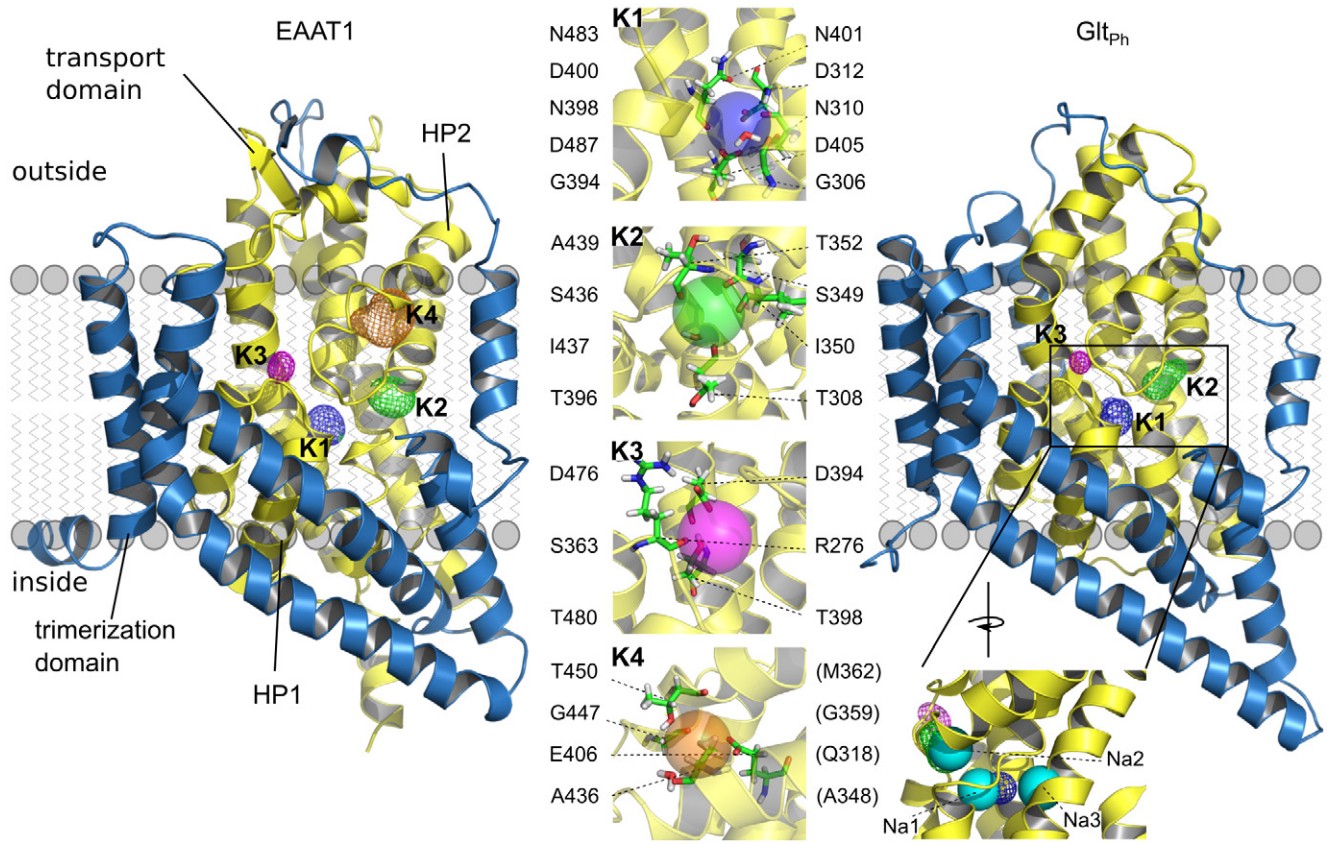

**Figure 1.  Unguided MD simulations identify K$^+$-binding sites in EAAT1 and Glt$_{Ph}$.**

Isodensity meshes illustrate the K$^+$ distribution (contoured at 3.5 σ) around EAAT1 and Glt$_{Ph}$ monomers in the OFC in side view. Middle insets show close-ups of the conserved K1–3 sites in Glt$_{Ph}$ and the K4 site in EAAT1 (left, coordinating EAAT1 residues; right, Glt$_{Ph}$ homologues). Lower right inset, comparison of the K1–K3 sites (isodensity meshes) with the Na$^+$-binding sites Na1–Na3 (spheres) (Boudker et al, 2007; Guskov et al, 2016). See also Appendix Figs S1–S3.

The low affinity and low selectivity exclude K3 and K4 as K$^+$-binding site during EAAT re-translocation.

K$^+$ binding to glutamate transporters results in transient capacitive currents (Grewer et al, 2012; Kovermann et al, 2017) caused by charge redistribution within the protein due to K$^+$ movement along the electric field, neutralization of the negatively charged binding sites, and partial translocation of charged protein domains across the membrane. Since K1–K4 sites assume separate positions within the electric field and their occupation results in distinct charge distributions within the transport domain, such capacitive currents permit assignment of the K$^+$-binding site occupied during translocation. Figure 3D depicts current responses in cells expressing human EAAT1 or EAAT2 upon rapid application of K$^+$ ions (Grewer et al, 2012). Transporters are predominantly in the OFC in the absence of external K$^+$ and isomerize to the IFC upon K$^+$ application generating a capacitive outward current. We simulated transmembrane voltages as a function of applied ionic charge imbalance for Glt$_{Ph}$ in outward- and inward-facing conformations, with K1, K2, K3, or K4 empty or occupied. Transmembrane voltages across the membrane/ protein capacitor depend on both ionic charge imbalances and protein charges, and such simulations therefore provide protein charge distributions and the voltage dependence of K$^+$ binding and K$^+$-bound translocation for K1–K4 (Fig 3E) (Machtens et al, 2017).

Our MD simulations reveal that inward translocation of the transport domain moves negative charges across the electric field and induces an outward current, when the K1 or the K4 site is occupied, in full agreement with the experimental results. Translocation after occupation of K2 generates opposite charge movement, whereas there is only negligible charge displacement with K$^+$ bound to K3 (Fig 3E).

K2, K3 and K4 exhibit affinities, selectivities or voltage dependencies of K$^+$ translocation that are inconsistent with experimental results, whereas K1 meets all expectations for the K$^+$-binding site occupied during EAAT re-translocation. To further validate the functional involvement of the K1 site, we identified mutations that alter K$^+$ binding to K1 via alchemical free-energy calculations and tested them experimentally. Simulations of OFC Glt$_{Ph}$ demonstrated that D405N and D312N reduced the free energy of K$^+$ binding to K1 by 84 and 40 kJ/mol, respectively, whereas N401A caused a reduction by only 12 kJ/mol (Fig 4A and Appendix Fig S5C). We measured K$^+$ binding to corresponding mutants of EAAT2 using whole-cell patch-clamp recordings. EAAT/Glt$_X$ can also function as anion channels that open from intermediate translocation states (Fig 4B, D and E; Fahlke et al, 2016; Machtens et al, 2015). EAAT anion channel activity is usually attributed to Na$^+$ and glutamate-bound states in the transport cycle (Vandenberg & Ryan, 2013; Zhou et al, 2014);

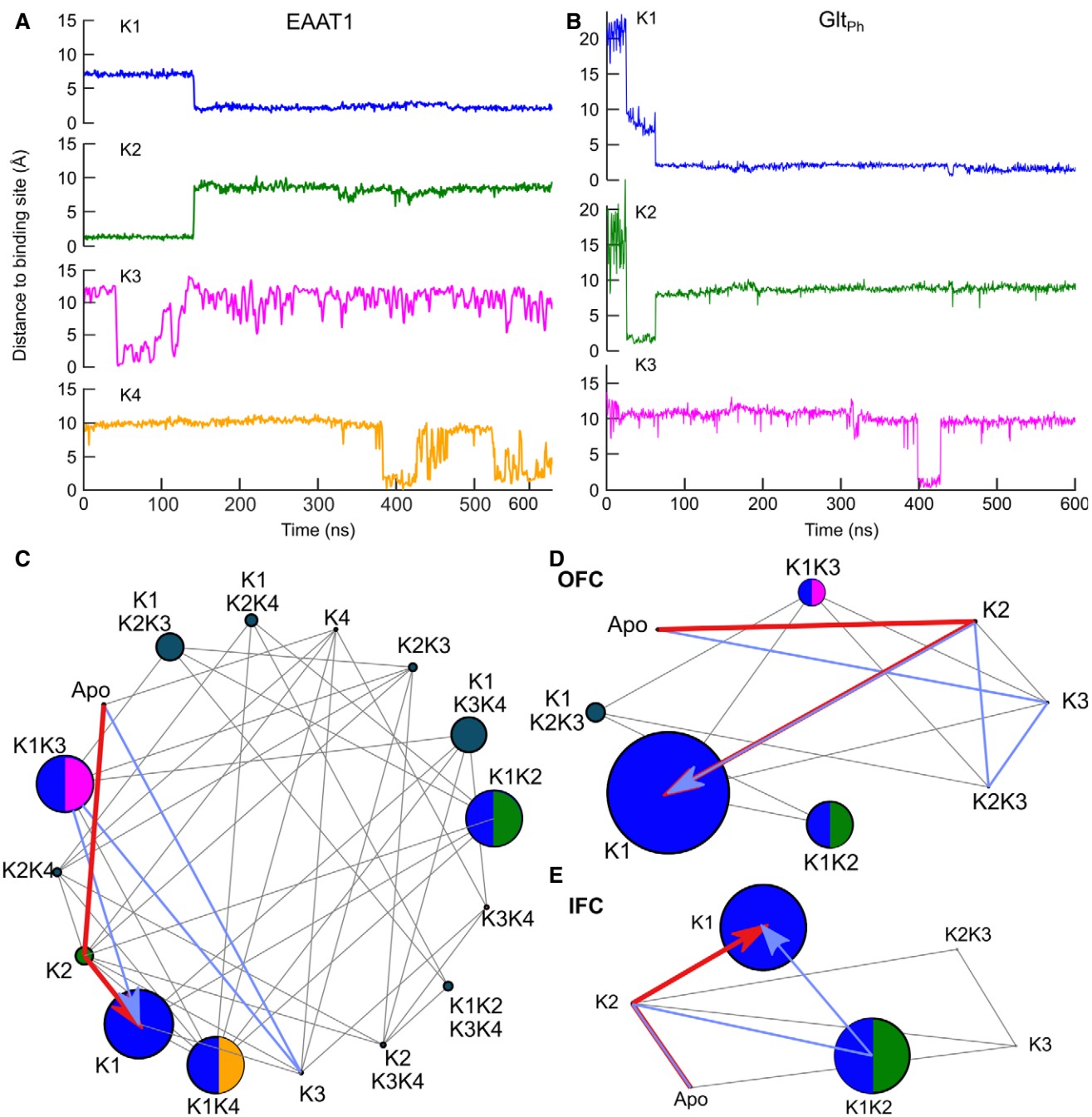

**Figure 2.  K⁺-binding path analysis demonstrates transient association to K2 and K3 preceding stable occupation of K1-bound states.**

A, B   Time course of representative binding/unbinding events in EAAT1 and Glt$_{Ph}$ simulations, shown as the distance of the closest K⁺ to the respective site.

C–E   Kinetic models of K⁺-binding steps for (C) EAAT1 and (D, E) Glt$_{Ph}$ (node radius, estimated equilibrium occupancy; lines, observed transitions; red and blue, two apo-K1 transition paths with the highest reactive flux).

Data information: See also Appendix Fig S4.

however, also K⁺ bound states exhibit anion channel activity. In EAAT2, anion currents are large with K⁺ as main internal and external cation, and the K⁺ dependence of anion currents can be used to measure the effect of mutations on the two hemicycles in isolation in a patch-clamp experiment (Fig 4B, D and E). D399N and D486N (Glt$_{Ph}$ D312 and D405) abolish activation of EAAT2 anion currents

by external K⁺. In contrast, N482A (Glt$_{Ph}$ N401) currents were increased by external K⁺, but significantly less by external Na⁺ and L-glutamate (Fig 4E) as expected due to the involvement of this residue in substrate interactions (Boudker *et al*, 2007). These results indicate that mutations impairing K⁺ binding to K1 abolish K⁺-bound translocation in EAATs.

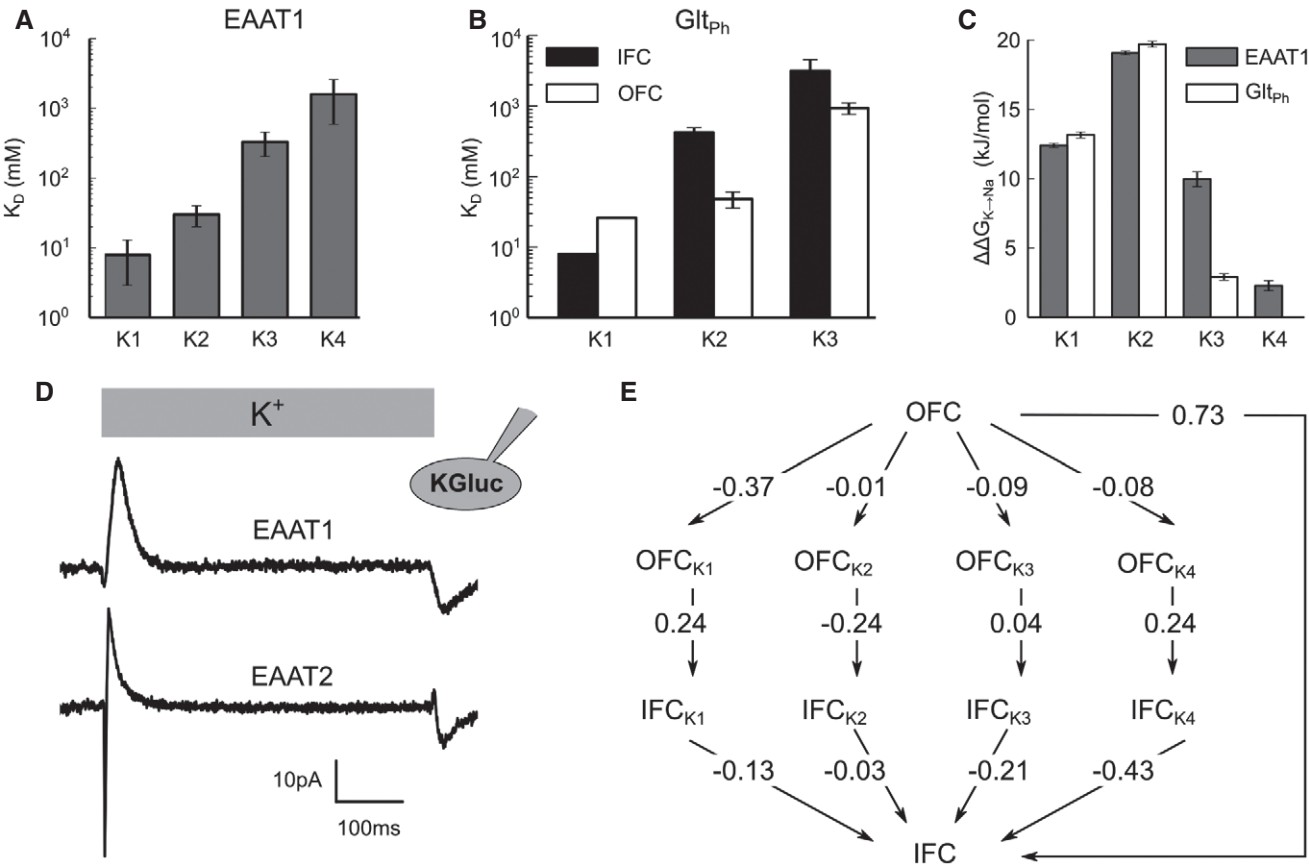

**Figure 3. K1 is the K$^+$-binding site responsible for K$^+$-dependent re-translocation.**

A, B   Dissociation constants for the K$^+$ sites in EAAT1 (A) and Glt$_{Ph}$ (B) from unguided MD simulations. Values are calculated by dividing off rates by second-order rate constants of ion binding. Mean ± SD of 1,000 bootstrap samples.

C   Na$^+$/K$^+$ selectivities of the K$^+$ sites in EAAT1 and Glt$_{Ph}$ (OFC) calculated from alchemical free-energy simulations. Mean ± SD of 1,000 bootstrap samples.

D   Mean current responses to rapid switching from 140 mM CholGluc to 140 mM KGluc of HEK293T cells expressing WT EAAT1 ($n$ = 3) or WT EAAT2 ($n$ = 3). Cells were intracellularly dialyzed with a 115 mM KGluc-based solution and held at 0 mV.

E   Simulated charge displacements associated with K$^+$ binding to outward, transporter translocation, and K$^+$ dissociation from inward-facing Glt$_{Ph}$ in units of $e_0$; SDs determined by bootstrap sampling range from $7 \times 10^{-3}$ to $8 \times 10^{-3}$ $e_0$.

Data information: See also Appendix Fig S5.

In radioactive assays, Glt$_{Ph}$ aspartate uptake is K$^+$ independent, and this result leads to the assumption that Glt$_{Ph}$ does not bind K$^+$ (Ryan *et al*, 2009). We tested K$^+$ binding to Glt$_{Ph}$ using microscale thermophoresis (MST), a technique that quantifies the thermophoretic mobility of fluorescently labeled proteins to detect K$^+$-dependent changes in protein charge or size. Glt$_{Ph}$ exhibits concentration-dependent changes in mobility (Fig 4C), in full agreement with conformational changes induced by K$^+$ binding. The D405N and the D312N mutations induce a right shift of the K$^+$ dependence in MST experiments (Fig 4C). Although D312N Glt$_{Ph}$ has impaired transport activity due to altered Na$^+$ interactions (Bastug *et al*, 2012), EAAT3 experiments indicate that this mutant still retains the ability to bind substrates and undergo substrate-induced conformational changes (Tao *et al*, 2006). Radioactive glutamate uptake experiments show that D405N Glt$_{Ph}$ is functional (Appendix Fig S6A). These results therefore confirm the importance of the K1 site for K$^+$ binding to Glt$_{Ph}$.

We next studied fluorescence quenching of a tryptophan (A233W) inserted into the transport domain of Glt$_{Ph}$ by spin-labeled 16-doxyl-stearate (16-SASL), a fatty acid with a quenching moiety near the membrane center (Subczynski *et al*, 2009). W233 resides at the lipid–water interface in the OFC and near the center of the membrane for transporters in the IFC (Appendix Fig S6B); motion of the transport domain toward the center of the lipid bilayer will thus reduce the fluorescence intensity. The tryptophan insertion does not impair glutamate uptake in Glt$_{Ph}$ (Appendix Fig S6A). TBOA traps Glt$_{Ph}$ in the OFC with open HP2 (Boudker *et al*, 2007; Ruan *et al*, 2017) and reduces fluorescence quenching (Appendix Fig S6C). Glt$_{Ph}$ translocation in the apo state increases quenching in choline-based solutions (Ryan *et al*, 2009), and application of KCl leads to an additional increase in quenching efficiency (Appendix Fig S6C). Constraining transporters in the inward-facing conformation by cysteine cross-linking of A233W K55C C321S A364C Glt$_{Ph}$ with Cu (II)(1,10-phenanthroline)$_3$ (Reyes *et al*, 2009; Ewers *et al*, 2013) maximally enhances quenching in KCl as well as in NaCl/TBOA. With Na$^+$ only, we observed intermediate fluorescence quenching of A233W Glt$_{Ph}$, likely due to stabilization of both the IFC and OFC.

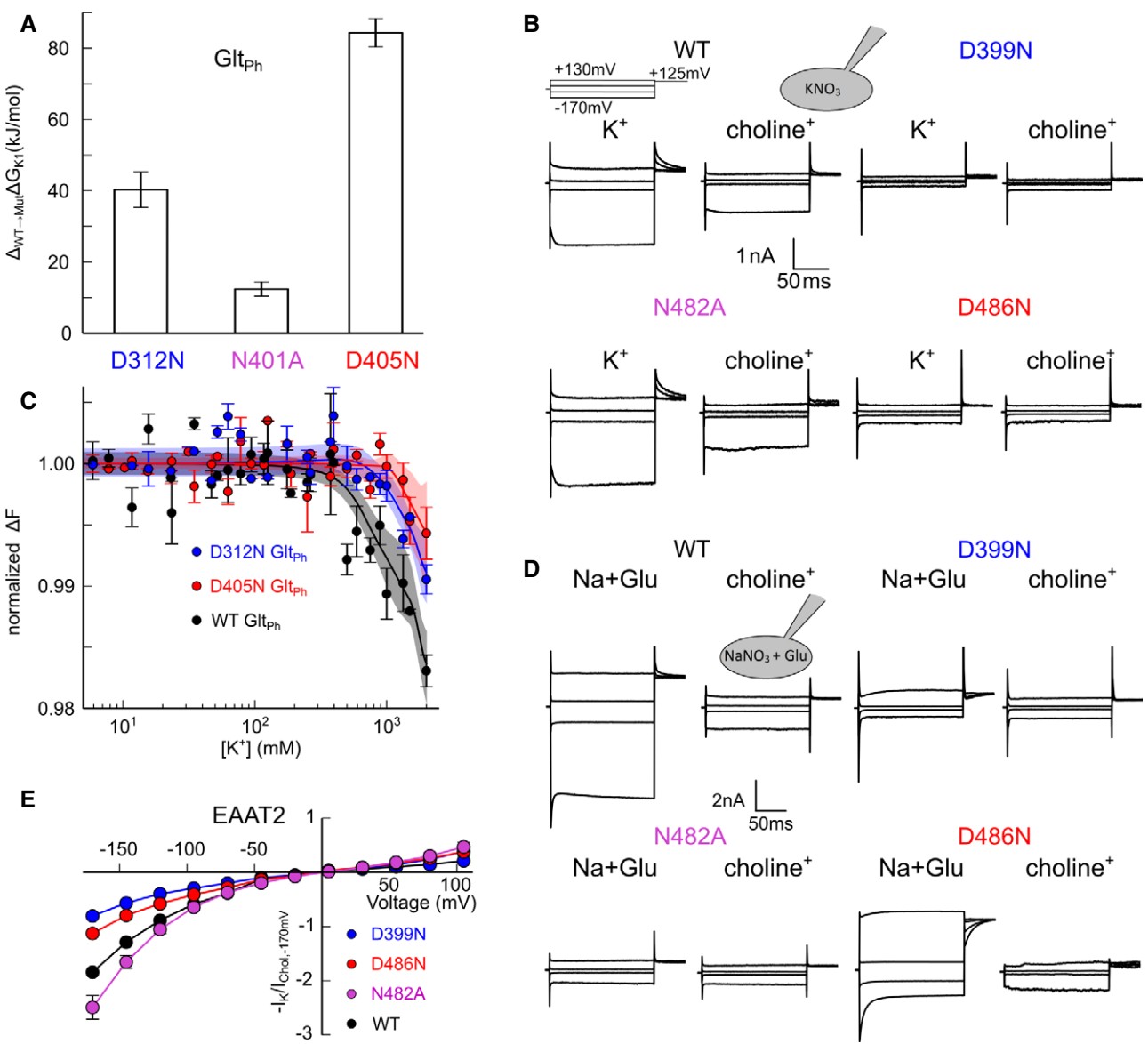

**Figure 4. K1 site mutations abolish K+ translocation in EAAT2 and K+ binding to GltPh.**

A   Simulated changes in the free energy of K+ binding to GltPh induced by different mutations. Values are calculated as differences of free-energy changes during alchemical transformations of amino acid side chains with the protein either in the K1 bound or in the apo state (Mean ± SD of 1,000 bootstrap samples).

B   Representative WT and mutant whole-cell current responses to voltage jumps before and after removal of external K+. Cells were dialyzed with a KNO3-based pipette solution and the external solution contained either 140 mM CholineNO3 or 140 mM KNO3.

C   Relative changes in fluorescence for WT, D312N, and D405N GltPh upon a temperature jump in MST experiments as a function of [K+]. Solid lines represent non-parametric fits, and shaded area shows bootstrapped 99%-confidence intervals for those fits. Non-overlapping confidence intervals indicate significant difference between WT and mutant binding curves. Values are given as mean ± SD ($n \geq 3$).

D   Representative whole-cell current recordings from cells internally dialyzed with NaNO3 and glutamate in choline-NO3-based external solutions or in NaNO3-based solutions supplemented with 1 mM L-glutamate. Glutamate-induced changes in anion current amplitudes indicate that D399N and D486N EAAT2 (GltPh D312 and D405) are K+-independent, but are functionally expressed in the plasma membrane.

E   Current–voltage relationships of steady-state currents from cells dialyzed with a KNO3-based pipette solution. Currents were measured at external 140 mM KNO3 and normalized to the current at −170 mV from a consecutive recording of the same cell in K+-free external solutions. Values are given as mean ± SD ($n \geq 3$).

Data information: See also Appendix Fig S6.

This is expected based on published single-molecule fluorescence resonance energy transfer (FRET) and EPR experiments (Erkens *et al*, 2013; Hänelt *et al*, 2013). Since average A233W quenching in the presence of Na+ (resulting in fractions of transporters kept in the inward or in the outward-facing conformations) is comparable with quenching in K+ (increasing the probability to translocate), the A233W quenching assay does not report on translocation on its own. This notwithstanding, these results provide further evidence

that $K^+$ binds to $Glt_{Ph}$. We conclude that EAATs and $Glt_X$ share the $K^+$-binding site K1, with binding affinity, selectivity, and voltage dependence of binding and translocation in agreement with experimental results. K2–K4 exhibit only low affinity or absent $K^+$ selectivity and must be regarded as transient interaction sites. The conservation of K1 between $Glt_X$ and EAATs indicates that development of $K^+$ coupling in the EAATs cannot be due to evolutionary creation of a novel cation-binding site.

### K1 binding closes the extracellular gate

Since translocation of the transport domain is only possible after closure of HP2 (Boudker *et al*, 2007; Vandenberg & Ryan, 2013; Verdon *et al*, 2014), we reasoned that differences in $K^+$ coupling between $Glt_X$ and mammalian EAATs might be due to distinct gate dynamics. In unguided simulations of EAAT1 OFC and $Glt_{Ph}$ OFC and IFC, we observed frequent spontaneous HP2 opening/closing (Fig 5A and B), but only subtle conformational changes of HP1 (Appendix Fig S7A), consistent with the notion that HP2 acts as the gate in both the OFC (Boudker *et al*, 2007; Verdon *et al*, 2014) and IFC (Zomot & Bahar, 2013). HP2 was mainly open in apo EAAT1; the presence of $K^+$ at either K1, K2, or K4 led to a population shift toward the HP2-closed state, while K3 occupancy opened HP2 (Fig 5A). In contrast, apo $Glt_{Ph}$ sampled both open and closed HP2 states equally well. In the IFC, open propensities were slightly lower, and $K^+$ binding at K1 or K2 stabilized the closed state even more effectively than in the OFC. K3 occupation opened HP2 in both conformations (Fig 5B). Thus, in both EAAT1 and $Glt_{Ph}$, $K^+$ binding at the K1 or K2 site increased the propensity of HP2 being closed.

We used umbrella sampling simulations to explore the underlying free-energy landscape of HP2 opening, which revealed a strong preference of apo EAAT1 for the open state, while open and closed states are both accessible for apo $Glt_{Ph}$ (Appendix Fig S7C). Integration of the calculated probability densities (Appendix Fig S7C)

confirms a significantly lower HP2 closed probability of apo EAAT1 than for $Glt_{Ph}$ (0.2% vs. 7.1%). For both EAAT1 and $Glt_{Ph}$, K1 occupation shifts the HP2 distribution toward the closed state, increasing the HP2 closed probability to 20 and 59%, respectively, thereby confirming the conclusions drawn from the unguided MD simulations (Fig 5A and B). Whereas HP2 can easily close without $K^+$ binding and permits $K^+$-independent transport by $Glt_{Ph}$, EAATs require $K^+$ binding to shut the gate for subsequent transmembrane translocation. $K^+$ binding to K1 induces a reversible conformational stretch of transmembrane helix 7 along the NMDGT motif—the signature sequence of glutamate transporters (Fig 5C and Appendix Fig S2). This conformational change causes multiple hydrogen bonds to form between TM7/TM8/HP1 and HP2, which stabilize the closed gate. This stabilizing effect is observed in both $Glt_{Ph}$ and EAAT1, but it is more pronounced in EAAT1 (Fig 5C; Appendix Fig S8).

### Stabilizing the closed gate renders EAATs $K^+$ independent

Our results indicate that the essential role of $K^+$ in EAAT re-translocation, but its optional role in $Glt_{Ph}$, is due to variation in HP2 dynamics. Differences in primary structure of $K^+$-coupled and uncoupled transporters might thus inform side chain substitutions that create $K^+$-independent EAATs by lowering the apo state gate open probability. EAAT1 simulations revealed substantial backbone flexibility of HP2 in the open state that was reduced upon HP2 closure (Fig 6A), indicating an entropic penalty that might counteract HP2 closure (Boudker *et al*, 2007; Verdon *et al*, 2014). This reduction in backbone flexibility upon EAAT1 gate closure was especially pronounced around L448 (Fig 6A and B). Sequence comparison revealed that L448 ($Glt_{Ph}$ A360) is fully conserved in EAATs, but not in $K^+$-independent transporters (Fig 6A and Appendix Fig S2).

We tested the consequences of amino acid substitutions at this position on HP2 dynamics in MD simulations (L448A and L448T)

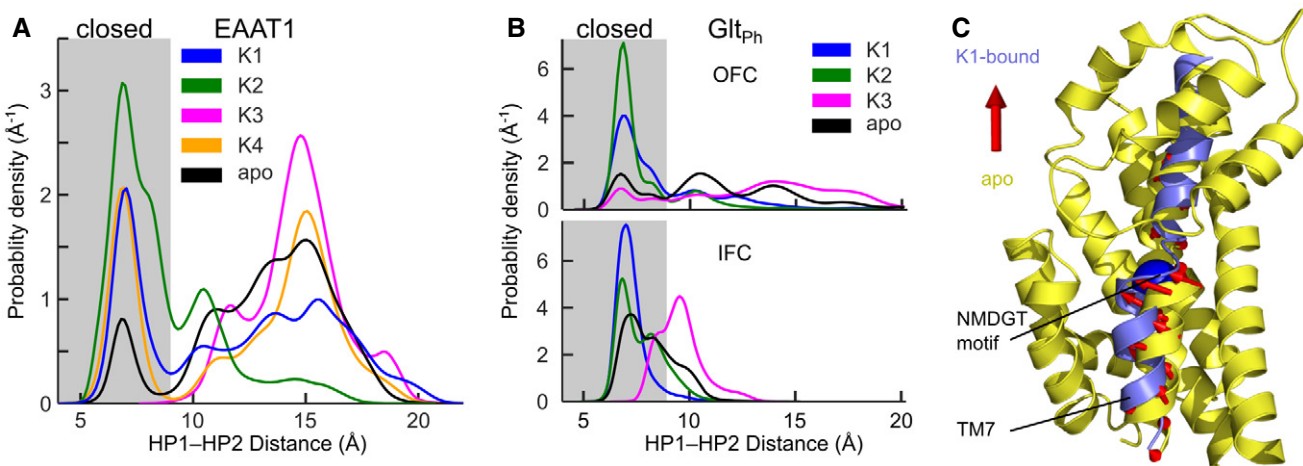

**Figure 5. K1 occupation leads to extracellular gate closure.**

A, B HP1–HP2 distance distributions for (A) EAAT1 (measured between residues 383 and 443) and (B) $Glt_{Ph}$ (measured between residues 276 and 355) obtained from unguided simulations as a function of $K^+$ occupancy. Inset, snapshots of the HP2 gate in open (gray) and closed (blue) states. The boundary between open and closed states was defined as the first local minimum of the K1-bound histogram.

C $K^+$-induced conformational changes of the EAAT1 transport domain. Overlay of TM7 before (yellow cartoon) and after (blue) K1 occupation.

Data information: See also Appendix Fig S7.

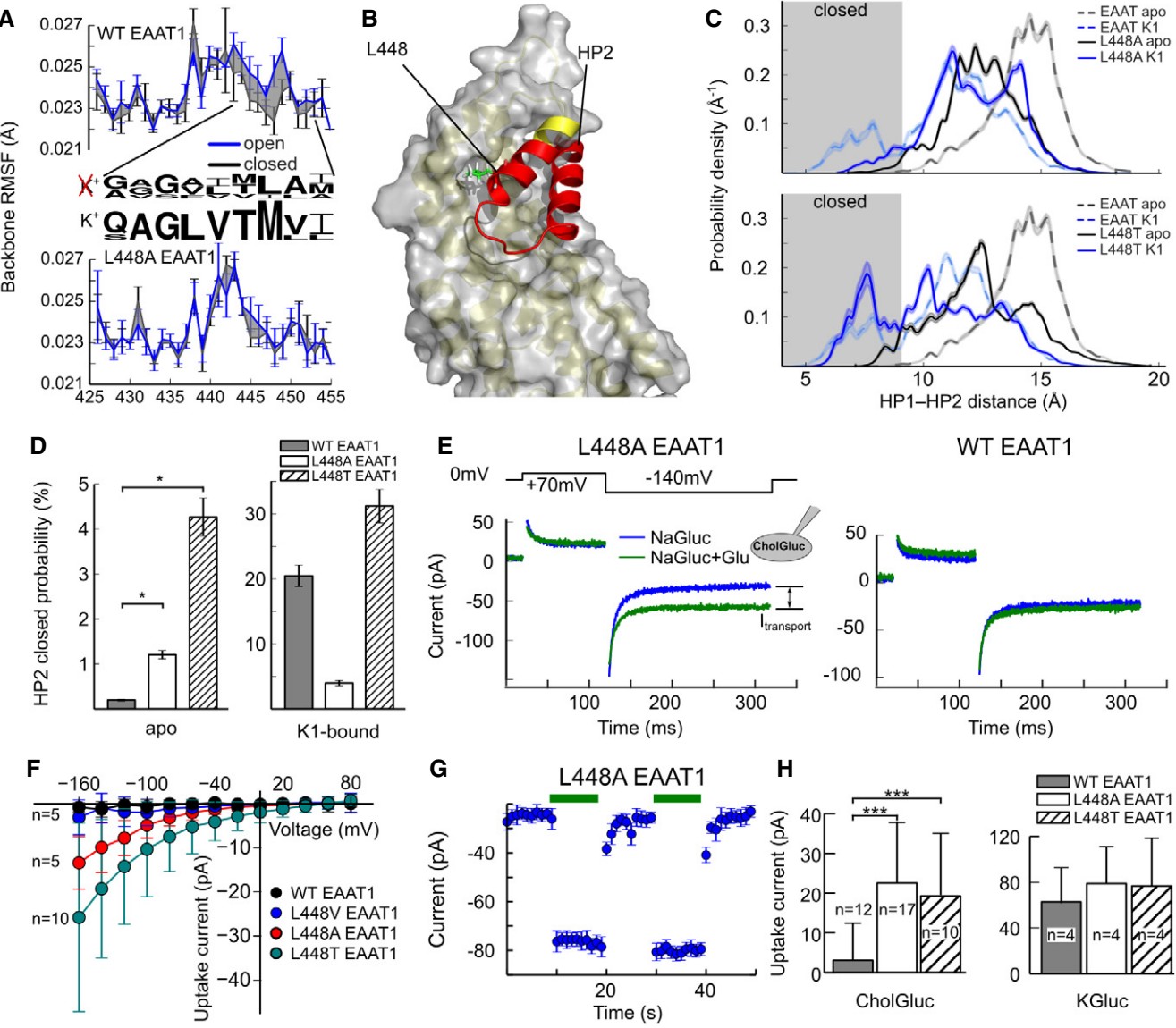

**Figure 6. Mutations that stabilize the closed gate generate K⁺ independent EAATs.**

A   Root-mean-squared fluctuation (RMSF) of the HP2 backbone in the open and closed state for WT and L448A EAAT1 (Glt$_{Ph}$ A360). Values are given as mean ± SD of 10 blocks. Amino acid conservation in the indicated stretch (residue range 444–452; Glt$_{Ph}$ 356–368) is shown as sequence logo for all K⁺-independent (Glt$_X$ and ASCT1/2, upper row) and all K⁺-dependent (EAAT1–5, lower row) transporters.

B   HP2 closure is accompanied by deformation of the backbone structure around residue L448 (Glt$_{Ph}$ A360).

C   Probability density distribution for HP2 opening in WT (from Appendix Fig S7C), L448A or L448T EAAT1 (Glt$_{Ph}$ A360); the shaded area indicates closed states.

D   Probabilities of HP2 closure from the integrated densities in (B) (mean ± SD from 1,000 bootstrap samples). Statistical significance was tested by comparing bootstrapped confidence intervals (*95%-confidence intervals do not overlap).

E   Representative transport current recordings from cells expressing L448A or L448T (Glt$_{Ph}$ A360) or WT EAAT1 upon voltage jumps to −140 mV before and after superfusion with 1 mM L-glutamate. Cells were intracellularly dialyzed with a K⁺-free solution without permeating anions to isolate K⁺-independent transport currents. Transient capacitive currents during the first 5 ms after a voltage jump were blanked.

F   Current–voltage relationship for net transport current amplitudes for WT and L448A/T EAAT1 (Glt$_{Ph}$ A360) for choline-based pipette solutions. Values are given as mean ± SD with indicated numbers of experiments.

G   Time-dependent changes in L448A EAAT1 (Glt$_{Ph}$ A360) uptake currents during two repetitive glutamate applications and subsequent removals. Current amplitudes were measured at the end of voltage jumps to −140 mV and plotted vs. the time. Glutamate application is indicated by green bars. Values are given as mean ± SD of the last 100 data points in the current trace.

H   Net transport current amplitudes for WT and L448A/T EAAT1 (Glt$_{Ph}$ A360) for different pipette solutions; values for internal KGluc are taken from current–voltage relationships in Appendix Fig S9. Values are given as mean ± SD with indicated numbers of experiments (***$P < 0.001$, Student's $t$-test).

Data information: See also Appendix Fig S9.

and on $K^+$-dependent transport in experiments (L448A, L448T, L448V). Free-energy calculations demonstrate that L448A and L448T (Glt$_{Ph}$ A360) render the closed state in apo EAAT1 energetically more favorable, resulting in an increased HP2 closed probability (Fig 6C and D). Additionally, L448A/T abolished the $K^+$ effect on EAAT1 HP2 closure (Fig 6D) by preventing hydrogen bonds between D476/R479 (Glt$_{Ph}$ D394/R397) and HP2 that would stabilize the closed gate in the $K^+$-bound state (Appendix Fig S8A).

To experimentally confirm that L448A/T confer $K^+$ independence, we measured transport currents in transfected cells under conditions that abolish anion currents (Wadiche *et al*, 1995). With choline$^+$ as main intracellular cation, L448A/T EAAT1 and homologous L447A EAAT2 showed glutamate-sensitive currents significantly larger than background currents, unlike wild-type (WT) EAATs (Fig 6E–H and Appendix Fig S9A). In contrast, the more conservative mutation L448V EAAT1 did not differ from WT in $K^+$ dependence. EAAT transport currents with $K^+$ as main cation were identical for WT and L448A/T EAAT1 and even larger for L447A EAAT2 than WT EAAT2, indicating that differences in $K^+$ independent transport current amplitudes are not due to differences in expression levels (Fig 6H). We tested multiple amino acid substitutions at other positions in HP2 differing between $K^+$-dependent and independent transporters (Fig 6A), but none of them resulted in $K^+$-independent EAATs (Appendix Table S6).

Current–voltage relationships (Fig 6G) demonstrate that we measured only inward transport currents in these experiments, without substantial contamination with additional EAAT1 current components, endogenous currents, or leakage currents. As additional test for $K^+$-independent transport in mutant EAATs, we studied the [$K^+$] dependence of glutamate transport current reversal potentials. We chose WT and L447A EAAT2 for these experiments because of their higher uptake current amplitudes. Cells were dialyzed with solutions containing Na$^+$, $K^+$, and glutamate and externally perfused similar with variable [$K^+$]. EAAT2 anion currents were abolished by complete anion substitution with the impermeant gluconate, and EAAT2 transport current reversal potentials were determined as TBOA-sensitive current responses to slow voltage ramps. Appendix Fig S9C depicts changes in transport current reversal potentials as a function of changes in external [$K^+$]. Whereas reversal potential shifts closely resemble the prediction for Na$^+$- and $K^+$-dependent glutamate transport (Zerangue & Kavanaugh, 1996) in WT EAAT2, results on L447A EAAT2 demonstrate a reduced $K^+$ dependence, in full agreement with the notion that mutant transporters can mediate $K^+$ coupled as well as $K^+$ independent transport.

Whereas EAATs transport glutamate together with Na$^+$ and H$^+$, in exchange with $K^+$, Glt$_{Ph}$ (Ryan *et al*, 2009) and Glt$_{Tk}$ (Arkhipova *et al*, 2019) are Na$^+$-aspartate symporters. This comparison suggests that H$^+$ and $K^+$ coupling has evolved simultaneously and may be functionally depended on each other. We therefore tested the pH dependence of L447A EAAT2 (Appendix Fig S9D and E) and found that glutamate transport remains H$^+$ dependent (Tao & Grewer, 2005).

Upon HP2 opening in EAAT1 simulations, we observed reversible salt bridge formation between E406 (TM7, Glt$_{Ph}$ Q318) and R479 (TM8, Glt$_{Ph}$ R397) that sterically blocked HP2 closure (Fig 7A and B). These residues are conserved in all $K^+$-coupled glutamate transporters (Appendix Fig S2), whereas the homologous residues Q318 and R397 did not interact in Glt$_{Ph}$ simulations (Appendix Fig S8F). Earlier work suggested that neutralization of this arginine

disrupts $K^+$ interactions with EAAT3 (Bendahan *et al*, 2000), and we hypothesized that dynamic formation of this salt bridge might contribute to the higher gate open probability in EAATs. Umbrella sampling simulations of R479A EAAT1—which cannot form the salt bridge—revealed an altered free-energy landscape, with lower HP2 open probability compared with WT (Fig 7C and D). The R479 side chain is part of an interaction network with D472 and D476 (Glt$_{Ph}$ D390 and D394); the R479A substitution disrupts the effect of K1 binding on this network, making interactions of D472 and D476 with HP2 independent from K1 occupation (Appendix Fig S8A). Since R479A also modifies the substrate selectivity (Bendahan *et al*, 2000), we measured the $K^+$ dependence of serine rather than of glutamate transport. Consistent with the MD simulations, we measured robust transport currents for R479A EAAT2 in the absence of intracellular $K^+$ (Fig 7E–G). L448A and R479A EAAT1 demonstrate that a $K^+$-independent transporter can be generated by shifting the equilibrium of the extracellular gate toward the closed state.

## Discussion

We here identify the mechanisms that couple glutamate transport to $K^+$ gradients in mammalian glutamate transporters, but allow for $K^+$-independent transport in prokaryotic homologs. We conducted extensive multi-µs MD simulations that identified three $K^+$-binding sites (K1–K3) in both Glt$_{Ph}$ and EAAT1 and the EAAT1-specific K4 site (Fig 1). Site geometries are similar in both IFCs and OFCs, as expected for elevator transporters (Reyes *et al*, 2009). K3 and K4 exhibit only low affinity and selectivity for $K^+$, with K3 occupation resulting in HP2 opening, i.e., preventing transport (Figs 3–5). K2 functions as a high-affinity $K^+$ scavenger that subsequently relays the $K^+$ ion to the K1 site (Fig 2). Only the K1 site provides both high affinity and sufficient $K^+$ selectivity as required for $K^+$-bound translocation in mammalian glutamate transporters. $K^+$ occupation of K1 induces HP2 closure to facilitate translocation and leads to charge redistribution in the transport domain consistent with recordings of $K^+$-induced currents (Fig 3). Mutations that impair $K^+$ binding to K1 abolish $K^+$-dependent changes in thermophoretic mobility in experiments with Glt$_{Ph}$ (Fig 4C) and $K^+$-dependent translocation in EAAT1 (Fig 4). The spatial proximity of K1 to the Na1 and Na3 sites ensures that Na$^+$ and $K^+$ binding are mutually exclusive, as required for a Na$^+$/$K^+$ exchanging transporter (Kanner & Bendahan, 1982).

$K^+$-induced gate closure in EAAT1 and Glt$_{Ph}$ is initiated by occupation of the K1 site, followed by an induced conformational stretch of the NMDGT motif in TM7, which recruits several hydrogen bonds with closed HP2 (Fig 5C; Appendix Fig S8). No experimental glutamate transporter structure with a bound $K^+$ ion was resolved so far, precluding a direct structural comparison with our simulations. However, recent crystal structures of the homologue Glt$_{Tk}$ have been resolved in an outward-facing, Na$^+$- and aspartate-free state, following Glt$_{Tk}$ protein purification in presence of [$K^+$] (Jensen *et al*, 2013; Guskov *et al*, 2016). Even though no $K^+$ density was assigned to a $K^+$ site, these structures might represent a $K^+$-bound conformation. Indeed, structural comparison with our simulations reveals that the structures of these $K^+$-exposed Glt$_{Tk}$ transporters resemble $K^+$-bound Glt$_{Ph}$ in our simulations. In particular, the

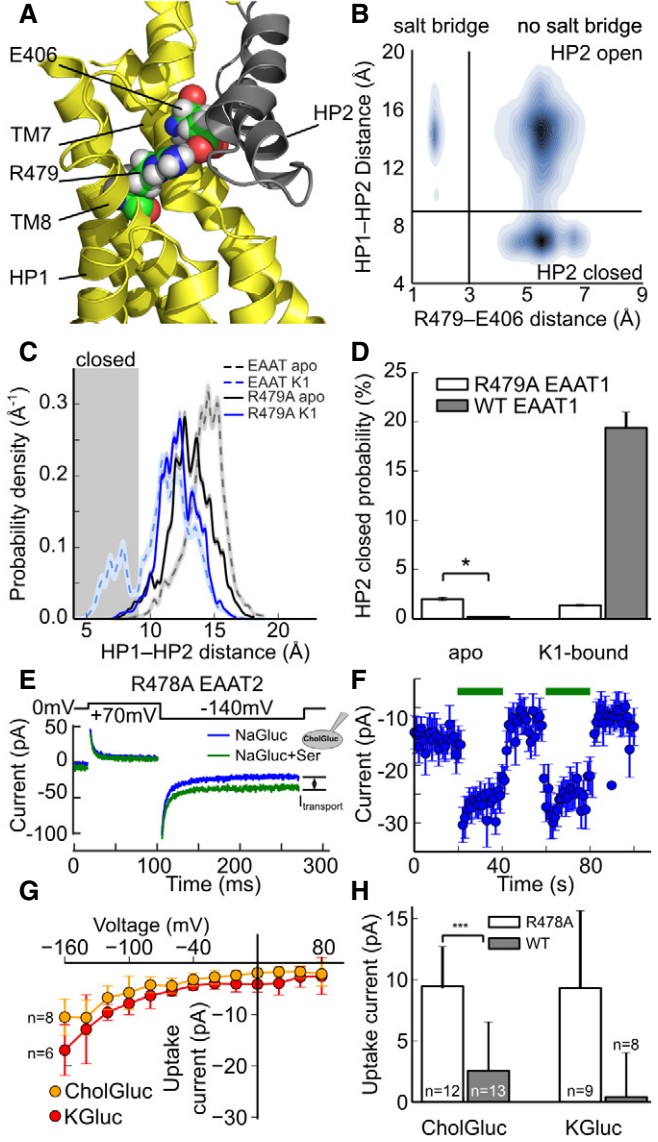

**Figure 7. A conserved salt bridge prevents HP2 closure in apo EAATs.**

A   The E406–R479 (Glt$_{Ph}$ Q318, R397) salt bridge locks HP2 in the open state in EAAT1. Image shows the region around residues 406 and 479 (shown as spheres) with HP2 open (gray cartoon). Both residues sterically prevent HP2 closure by forming a salt bridge.

B   Contour plot of the HP distance vs. the R479–E406 (Glt$_{Ph}$ Q318, R397) distance distribution. Gray lines indicate thresholds for the HP2-closed and salt bridge-formed states.

C   Probability density distribution for HP2 opening in WT (from Appendix Fig S7C) and R479A EAAT1 (Glt$_{Ph}$ R397).

D   Probability of HP2 closure calculated by integrating the densities in (C) (mean ± SD from 1,000 bootstrap samples). Statistical significance was tested by comparing bootstrapped confidence intervals (*95%-confidence intervals do not overlap).

E   Representative transport current recordings from a cell expressing R478A EAAT2 (Glt$_{Ph}$ R397) in the absence of internal K⁺. Transport currents were elicited by voltage jumps to −140 mV before and after superfusion with 5 mM ʟ-serine. Transient capacitive currents during the first 5 ms after a voltage jump were blanked.

F   Time-dependent changes in steady-state R478A EAAT2 (Glt$_{Ph}$ R397) uptake currents during two consecutive serine applications and subsequent removals. Current amplitudes were measured at the end of voltage jumps to −140 mV and plotted vs. time. Serine application is indicated by green bars. Values are given as mean ± SD of the last 100 data points in the current trace.

G   Current–voltage relationship for mean net transport currents for R478A EAAT2 induced by 5 mM ʟ-serine for different intracellular solutions. Values are given as mean ± SD for the indicated number of experiments.

H   Mean net transport currents for WT and R478A EAAT2 (Glt$_{Ph}$ R397) induced by 5 mM ʟ-serine for different intracellular solutions. Values are given as mean ± SD for the indicated number of experiments.

During physiological glutamate uptake, EAATs bind K⁺ in the IFC and then isomerize to the OFC, from which K⁺ dissociates into the external solution. The limited availability of structural information restricted our EAAT1 simulations of K⁺ binding/unbinding to the OFC (Canul-Tec *et al*, 2017); however, all available simulations and experimental results demonstrate that the identified K⁺-coupling mechanism accounts for forward and reverse glutamate transport. First, EAATs efficiently transport in both forward and reverse directions with identical coupling stoichiometry (Zerangue & Kavanaugh, 1996). Second, although the details of ion/substrate binding to the inward-facing transporter are not fully understood, molecular simulations suggest that HP2 can act as a gate for translocation not only in the OFC, but also in the IFC (Zomot & Bahar, 2013 and Appendix Fig S7). Third, the interactions formed between HP2 and the transport domain are identical in OFC and IFC Glt$_{Ph}$ (Appendix Fig S7), and Glt$_{Ph}$ simulations show that K⁺ binding exerts similar effects on HP2 dynamics in both conformations (Fig 5B). Finally, we identified EAAT1/2 mutations that permit K⁺-independent forward transport based on our analysis of HP2 dynamics in the OFC (Figs 6 and 7).

EAAT K⁺-binding sites have been sought for decades. A recent MD study by Heinzelmann *et al* proposed Na1 as a K⁺-binding site using a Glt$_{Ph}$-based EAAT3 homology model (Heinzelmann & Kuyucak, 2014). They performed alchemical free-energy calculations on the OFC with closed HP2 that did not consider K⁺-binding HP2 coupling and thus predicted K⁺ affinities significantly above experimental values (Heinzelmann & Kuyucak, 2014). Na1 differs from the here-described K1 site in lacking coordination by the β-carboxylate of D400 (Glt$_{Ph}$ D312) and is only negligibly K⁺/Na⁺ selective

simulated K⁺-induced conformational change in the NMDGT motif is similar to the conformational difference between the Glt$_{Ph}$ structure obtained in K⁺-,Na⁺-, and aspartate-free conditions (PDB ID: 4OYE) and the structures of K-exposed Glt$_{Tk}$ (PDB IDs: 4KY0, 5DWY; Appendix Fig S8E).

Molecular simulations of K⁺ binding to EAAT1 are based on the recently determined crystal structure of an engineered Na⁺- and aspartate-bound EAAT1 (Canul-Tec *et al*, 2017). This protein contains several thermostabilizing mutations, some of which might cause slight structural deviations from the WT structure. However, the thermostabilized transporter exhibits robust Na⁺- and K⁺-dependent glutamate uptake, indicating that the mechanisms of Na⁺ and K⁺ coupling are preserved (Canul-Tec *et al*, 2017). Moreover, simulations of EAAT1 and Glt$_{Ph}$ provided consistent results about K⁺-binding sites, the effects of K⁺ binding as well as on HP2 dynamic. We are therefore confident that the use of the thermostabilized EAAT1 does not affect any of our conclusions.

(Heinzelmann & Kuyucak, 2014). The lack of K$^+$ selectivity strongly argues against Na1 as K$^+$-binding site. Holley and Kavanaugh used valence shell calculations on an EAAT3 homology model to identify various alkali cation-binding sites (Holley & Kavanaugh, 2009). Their approach did not consider protein flexibility and hydration effects and consequently missed the K1 site, but identified a site resembling K3 (Holley & Kavanaugh, 2009). In agreement with our findings, the authors reported imperfect K$^+$ to Na$^+$ selectivity and preferred occupation of this site with open HP2 (Holley & Kavanaugh, 2009). Experimental evidence for K3 as K$^+$-binding site is scant: thallium soaking of apo Glt$_{Ph}$ crystals with closed HP2 yields densities at a site similar to K3 (Verdon et al, 2014), but K$^+$ competition experiments failed to demonstrate K$^+$ binding at this site (Verdon et al, 2014). Mutations affecting the K3 site (Ryan et al, 2010; Wang et al, 2013) resulted in changes in the apparent substrate and K$^+$ dependence of transport and anion currents, but failed to unambiguously demonstrate K$^+$ binding to K3 during translocation. Our simulations reveal low K$^+$ affinity and selectivity, a voltage sensitivity of K3 occupation and translocation inconsistent with electrophysiological recordings, and finally K3 occupation-induced HP2 opening, thus excluding K3 as functional K$^+$-binding site.

K4 is only present in EAAT1, but not in Glt$_{Ph}$, and might thus contribute to isoform-specific differences in K$^+$ coupling. Substitution of E404 EAAT2 (Glt$_{Ph}$ Q318), which is part of K4, converted EAAT2 into a glutamate homoexchanger (Kavanaugh et al, 1997), but mutations of this residue were later shown not to prevent K$^+$-coupled transport but H$^+$ coupling instead (Grewer et al, 2003). In our simulation, K4 is characterized by extremely low affinity and selectivity, thus being unable to serve as K$^+$-binding site during translocation. We consider the observed K$^+$ density at K4 unspecific cation binding to the established H$^+$ acceptor E406 (Grewer et al, 2003).

By simultaneously monitoring K$^+$ binding and HP2 dynamics via extensive unguided MD simulations, our study unequivocally converges on the K$^+$-selective K1 site, in good agreement with previous and our own mutagenesis experiments and measured affinities (Appendix Figs S4 and S5; Mwaura et al, 2012; Rosental et al, 2006; Teichman et al, 2009; Zerangue & Kavanaugh, 1996).

K1 is also present in Glt$_{Ph}$, and MST experiments (Fig 4) and fluorescence quenching experiments (Appendix Fig S6) confirm K$^+$ binding to the K1 site in Glt$_{Ph}$. However, we cannot draw definite conclusions about the functional significance of K$^+$ binding in Glt$_{Ph}$. Aspartate-bound translocation is slow and rate-limiting in Glt$_{Ph}$ (Akyuz et al, 2013; Ruan et al, 2017), and K$^+$ binding might thus accelerate re-translocation without affecting the overall transport rate. While previous studies on Glt$_{Ph}$ were performed with low [Na$^+$] (Ryan et al, 2009; Ewers et al, 2013), its host organism *Pyrococcus horikoshii* thrives at 97°C in 400–500 mM NaCl (Gonzalez et al, 1998), and Glt$_{Ph}$ will thus experience significantly higher cytosolic NaCl concentration. These extremely denaturing conditions require outstanding thermostability of Glt$_{Ph}$, and K$^+$ might serve as co-factor to increase thermostability (Epand et al, 1999).

EAAT/Glt$_X$ translocation requires a compact shape of the transport domain with closed HP2 to prevent steric clashes with the trimerization domain (Verdon et al, 2014). The high probability of HP2 closure without K$^+$ binding (77% of the time in unbiased simulations for the apo IFC) permits effective K$^+$-independent re-translocation in Glt$_{Ph}$. In EAAT1, a modified free-energy landscape enables

HP2 closure and efficient inward/outward translocation of the transport domain only upon K1 occupation (Fig 5A and B, and Appendix Fig S7C). K$^+$ binding allosterically shifts the equilibrium of HP2 toward the closed state, which can be formally described as K-type allostery (LeVine et al, 2016). Furthermore, under non-equilibrium conditions, the velocity of elevator translocation limits the overall transport rate. For K$^+$-induced HP2 closure, the K-type allosteric efficacy α can be calculated as:

$$\alpha_{\text{HP2 close}}^{\text{K+ bind}} = \frac{K_{\text{HP2 close}}^{\text{K}^+}}{K_{\text{HP2 close}}^{\text{apo}}}$$

where $K_{\text{HP2 close}}$ is the equilibrium constant for HP2 closure in absence or presence of a K$^+$ ion. WT EAAT1 exhibits much higher allosteric efficacies (α = 114) than Glt$_{Ph}$ (α = 20). K$^+$-uncoupled EAAT1 mutants show reduced efficacies (L448A: α = 6; R479A: α = 0.7). Tuning allosteric K$^+$–HP2 coupling thus alters the K$^+$ coupling stoichiometry in glutamate transporters.

Our findings highlight the key role of allostery in the evolutionary optimization of transport stoichiometry in elevator transporters: Whereas in other transporter families the addition of a novel substrate to the transport cycle requires either the formation of a novel binding site or modification of their chemistry (Forrest et al, 2007; Zomot et al, 2007), the spatial separation of the control of the transport stoichiometry and the substrate-binding sites facilitates the evolutionary optimization of the transport stoichiometry without interfering with substrate binding. In enzymes or ion channels, protein function is modified by altering the dynamics of protein domains by phosphorylation, protonation, second messenger binding, or changes in the lipid environment. We speculate that transient modification of regulatory components of transporter translocation might also switch the cotransport of selected substrates on and off in elevator-type transporters, thus enabling adjustment of the transport direction to cellular needs in response to cellular signaling pathways.

The allosteric coupling mechanism provides a unifying molecular mechanism for stoichiometric coupling of K$^+$ and L-glutamate transport in mammalian EAATs. It reveals a new paradigm for fine-tuning the stoichiometry of secondary active transporters and might help in designing therapeutic approaches to block/enhance glutamate transport or even to modify substrate and voltage dependence. Such novel tools could also be useful for preventing reverse glutamate transport, a key mechanism underlying cell death in brain ischemia (Rossi et al, 2000), and for developing treatments for neurological diseases associated with increased neurotoxic, external glutamate concentrations.

# Material and Methods

### MD simulations

All MD simulations were performed using GROMACS 5 (Abraham et al, 2015), with the AMBER99SB-ILDN force field for the protein, Joung parameters for the ions (Joung & Cheatham, 2008), and the SPC/E water model. Glt$_{Ph}$ (PDB IDs: 2NWX, 4OYE, 3KBC) was modeled using the residue range 6–416 (Boudker et al, 2007; Reyes et al, 2009; Verdon et al, 2014), and the human EAAT1cryst-II construct (PDB ID: 5LLU) was modeled using the residue range

39–490 (Canul-Tec *et al*, 2017; corresponding to residues 39–510 in WT EAAT1; all reported EAAT1 residues use WT numbering). We used Glt$_{Ph}$ rather than Glt$_{Tk}$ as model system for K$^+$-uncoupled Glt$_X$ transporters, since more crystallographic (inward- and outward-facing states) and more functional information is available. For simulations on outward-facing Glt$_{Ph}$, we removed all bound ligands (Na$^+$ and aspartate) from the original fully bound Glt$_{Ph}$ structures (PDB IDs: 2NWX and 3KBC) and added missing atoms or residues using MODELLER (Webb & Sali, 2014). Fully bound Glt$_{Ph}$ structures exhibit higher resolution than available Glt$_{Ph}$ structures obtained in Na$^+$-only or apo states (Verdon *et al*, 2014) and lack engineered point mutations in the substrate-binding pocket. KCl was added to obtain bulk concentrations from < 1 mM up to 1 M (Appendix Table S1). A pressure of 1 atm and a temperature of 310K were attained using the Parrinello-Rahman barostat, and the velocity-rescale thermostat, respectively. *g_membed* (Wolf *et al*, 2010) was used to insert trimeric Glt$_{Ph}$ and EAAT1 into a pre-equilibrated POPC bilayer surrounded by an aqueous KCl solution, as informed by the OPM (Orientations of Proteins in Membranes) database (Lomize *et al*, 2006). Structures were initially equilibrated for 4 ns with position restraints on the protein heavy atoms and the z-component of lipid molecules, followed by 500–700 ns with position restraints on the protein heavy atoms only to ensure optimal equilibration of the lipid membrane around the experimental protein structure, and then 8 ns with backbone-only restraints to equilibrate the side chains. Starting from equilibrated systems, multiple independent microsecond-long MD simulations were initiated to study spontaneous K$^+$ binding and unbinding (Appendix Table S1). To simulate the protein in its apo state, we used a reduced K$^+$ concentration, where the simulation box contained only 6 K$^+$ ions (Figs 5–7). Ion permeation or electroporation events were not observed in any simulation within this study.

## K$^+$-binding analysis

We identified K$^+$ binding sites by analyzing the K$^+$ density distributions in free MD simulations. For each site, the mean squared distance to the coordinating atoms was calculated for each K$^+$, and distance trajectories were discretized into bound or unbound states using a distance threshold of 3.3–3.65 Å separately for each monomer. We then generated a time series of discrete K$^+$-occupation states ($16 = 2^4$ states for EAAT1, $8 = 2^3$ states for Glt$_{Ph}$) that were subsequently used for calculating the dwell times for each state (Appendix Tables S4 and S5) and the number of transitions between states (Appendix Tables S2 and S3). Transition rates between states were determined using equation (1) representing the maximum-likelihood estimate for chemical reaction rates (Prinz *et al*, 2011).

$$k_{01} = \frac{N_{01}}{t_0} \tag{1}$$

with $N_{01}$ being the accumulated observed number of transitions from state 0 to state 1 and $t_0$ the accumulated total dwell time of state 0. Equilibrium constants were calculated by dividing off rates by on rates. Since K1 dissociation was never observed in Glt$_{Ph}$, we decided to estimate an upper limit to the $K_D$ for the K1 site by assuming one K1–K2 relocation event in the total simulated dwell

time in the K1-bound states for Glt$_{Ph}$ (Appendix Fig S3F). Based on the determined $K_D$ for the K2 site, we estimated the $K_D$ of the K1 site to be < 26 mM in Glt$_{Ph}$. Transition path analysis was performed on this kinetic model using MSMExplorer (Hernández *et al*, 2017).

## Alchemical free-energy calculations

Na$^+$/K$^+$ selectivity of each binding site was determined using alchemical free-energy calculations (Appendix Fig S5). The change in free energy along the reaction between a K$^+$-bound state with a Na$^+$ ion in solution to a Na$^+$-bound state with a K$^+$ ion in solution was calculated from work distributions obtained from fast-switching simulations, in which the ions were transformed using GROMACS topologies generated with *pmx* (Gapsys *et al*, 2015). Harmonic position restraints were applied to all protein backbone atoms and to the coordinating side chains to avoid rearrangement of coordinating residues or relocation of ions between neighboring sites, for example of Na$^+$ from K1 to neighboring Na1 or Na3 sites. Reference geometries for the restraints were identified based on unbiased microsecond-long MD trajectories via *k*-means clustering of the K$^+$ site geometries in presence of a bound K$^+$.

To predict the effect of amino acid exchanges on K$^+$ binding, we calculated changes in free energy upon alchemical transformation of an amino acid side chain in a K$^+$-bound or in an apo monomer (Appendix Fig S6). To maintain neutrality of the simulation system during charge altering mutations, we performed the reverse transformation in an adjacent K$^+$-free monomer of the Glt$_{Ph}$ trimer, with one protomer in the apo state and another bound by one K$^+$ ion at the K$^+$-binding site. Monomers function independently (Grewer *et al*, 2005), and $\Delta G2$ and $\Delta G4$ values thus report K$^+$ affinity of the WT and mutant monomer, independently of modifications in adjacent monomers.

For both the Na$^+$/K$^+$ and side chain alchemical conversions, unbiased MD simulations were initiated from each of the two respective end states. We extracted ~ 250 snapshots from these trajectories (separated by intervals of 800 ps) and used them as starting structures for fast-switching simulations, in which the side chain was alchemically transformed within 5 or 10 ns, respectively. The first 40 ns of the end state equilibrium simulations were discarded for equilibration. Resulting work distributions were analyzed with the Crooks Gaussian intersection method (Goette & Grubmüller, 2009), as implemented in *pmx* (Gapsys *et al*, 2015). Crooks' fluctuation theorem (Crooks, 1998) relates the work distributions of forward- and backward-switching to the free-energy change between two states:

$$P_{AB}(W) = P_{BA}(-W)e^{\beta(W - \Delta G_{AB})} \tag{2}$$

where $P_{AB}$ and $P_{BA}$ denote the work distributions of the forward- and backward-switching processes, respectively. As a direct consequence, the free-energy change between states $A$ and $B$ is the work value at which both distributions intersect:

$$W = \Delta G_{AB} \Leftrightarrow e^{\beta(W - \Delta G_{AB})} = 1 \Leftrightarrow P_{AB}(W) = P_{BA}(-W) \tag{3}$$

Convergence of the simulations was assessed by dividing the data into two blocks and comparing the block-wise results. Simulations

were considered converged, when $\Delta G_{AB}$ estimates from the two blocks differed < 10%.

## Umbrella sampling simulations

We used principal component analysis (PCA) to define a single reaction coordinate, essential dynamics (ED) sampling (Amadei *et al*, 1996) to sample along this coordinate and umbrella sampling to quantify the free-energy profile of HP2 opening. A trajectory containing multiple microsecond-long free MD trajectories (initiated from apo, substrate-, or TBOA-bound Glt$_{Ph}$ after removal of TBOA, or EAAT1 conformations) were constructed and used for a cartesian PCA. The covariance matrix for the positional fluctuations of the backbone atoms of the transport domain (excluding loops) in the transporter monomer was calculated and diagonalized. The resulting first eigenvector was dominant with an eigenvalue exceeding others at least by a factor of three and was used as reaction coordinate. Essential dynamics was used to simulate the full HP2 opening transition by increasing the distance along this eigenvector during the MD simulation. A fixed increment linear expansion rate was chosen to simulate HP2 opening within 1.3 μs for Glt$_{Ph}$ and 650 ns for EAAT1. Windows for umbrella sampling were constructed from snapshots of the ED simulations, and the same eigenvector served as reaction coordinate, along which 44 (49 for EAAT1 K1) windows with structures from the ED simulation were used for umbrella sampling in which a harmonic potential restrained the projection on the first eigenvector. Each window was initially equilibrated using the Berendsen barostat; each production run was then simulated for 100 ns with a force constant of 1,000 kJ/mol/nm$^2$, the first 10 ns being discarded for equilibration. The resulting probability distributions were estimated using the weighted histogram analysis method in g_wham and assessed for statistical errors with bootstrap sampling (Hub *et al*, 2010). Convergence was tested by dividing the data into 10 ns blocks and comparing with the profile generated from the full data. Simulations were considered converged when the probability profiles from the last three blocks showed an overlap, of at least 80%.

## Calculation of charge displacement

We calculated the net charge transfer across the membrane associated with K$^+$ binding following a recently established approach for quantifying charge displacement in ion channels and transporters (Machtens *et al*, 2017). Computational electrophysiology setups (Kutzner *et al*, 2016), using anti-parallel double-bilayer orientation, were constructed, in which the protein was restrained to a single conformation. These systems were simulated for 60 ns under different ionic charge imbalances, and the resulting transmembrane voltages were calculated from the charge densities using Poisson's equation. In this setup, the membrane voltage, $V$, is given by equation (4):

$$V = \frac{q}{C_0} = \frac{(q_{sol} + q_{p0})}{C_0} \tag{4}$$

with $q_{sol}$ representing the ionic charge imbalance in the aqueous compartment, $q_{p0}$ the contribution of the protein to the total capacitor charge in state 0, and $C_0$ the total membrane/protein capacitance.

We measured the voltage in the system upon variation of $q_{sol}$ and determined $q_{p0}$ by linear regression. With imposed ionic charge imbalances in the range of $-12$ to $-6$ e$_0$, we obtained transmembrane voltages in the range of $-300$ to 300 mV, following a perfect linear relationship (Machtens *et al*, 2017). The $q_{p0}$ value depends on the conformation of the protein, and differences between protein conformations were used to calculate the effective charge movement between the conformations (Fig 3E). The applied voltages did induce neither ion permeation nor electroporation.

## Functional characterization of WT and mutant EAATs in mammalian cells

WT and mutant EAAT1/EAAT2 were expressed as fluorescent fusion proteins by transient transfection of HEK293T cells, as described previously (Machtens *et al*, 2011). Point mutations were introduced using overlapping extension PCR. All constructs were verified by restriction analysis and DNA sequencing, and two independent clones from the same transformation were checked to confirm identical function for each construct. Standard whole-cell patch-clamp recordings were performed using an EPC10 amplifier (HEKA Elektronik, Lambrecht, Germany). Currents were filtered at 10 kHz and sampled at 50 kHz. Borosilicate pipettes were pulled with resistances between 1.0 and 3.5 MΩ, and voltage errors were reduced by compensating 80–90% of the series resistance by an analogue procedure and excluding cells with current amplitudes higher than 12 nA from analysis. The standard bath solution for patch-clamp experiments contained (in mM) 140 NaNO$_3$, 1 MgCl$_2$, 2 CaCl$_2$, 5 TEA-Cl, 10 HEPES, adjusted to pH 7.4 with NMDG. In some experiments, 5 mM L-glutamate was added to the standard solution or NaNO$_3$ was substituted by equimolar KNO$_3$ or choline-NO$_3$. Pipette solutions contained either buffer A (115 KNO$_3$, 2 MgCl$_2$, 5 EGTA, 10 HEPES, adjusted to pH 7.4 with KOH) or buffer B (115 NaNO$_3$, 5 L-glutamate, 2 MgCl$_2$, 5 EGTA, 10 HEPES, adjusted to pH 7.4 with KOH). The external solution was exchanged by moving the cell into streams of different solution that were applied with a homemade gravity-driven perfusion system.

To measure electrogenic glutamate transport, nitrate was replaced by gluconate in both bath and pipette solutions. Steady-state transport currents were measured at $-140$ mV by subtracting currents measured in Na-gluconate from currents measured in Na-gluconate supplemented with 1 mM L-glutamate. To quantify K$^+$-independent uptake currents, KNO$_3$ in the pipette was replaced by choline gluconate (adjusted to pH 7.4 with choline-OH) and KCl-based agar bridges were replaced by NaCl-based agar bridges. Fast solution exchange experiments were done using a theta-glass mounted to a piezo device (Siskiyou Corporation, Grants Pass, Oregon). Currents were corrected for the background measured in solutions supplemented with 4 mM Na-gluconate and 2 μM TFB-TBOA. In experiments with K$^+$-free pipette solutions, cytoplasmic and pipette [K$^+$] will adjust with time constants that depend on series resistances and cell sizes (Pusch & Neher, 1988). In these experiments, mean series resistances measured 5.2 ± 1.9 MΩ ($n = 57$) and mean capacitances 20.7 ± 7.5 pF, corresponding to a concentration adjustment time constant of about 15 s. This analysis predicts that intracellular [K$^+$] will fall below 1 μM within 75 s assuming 150 mM K$^+$ after establishing the whole-cell mode with a

$K^+$-free pipette solution. We regularly waited at least 3 min after opening the cell before starting the experiments.

### Expression and purification of Glt$_{Ph}$

WT and mutant Glt$_{Ph}$ with a C-terminal 8× histidine tag cloned into a pBAD24 vector (provided by Dr. Eric Gouaux, Oregon Health and Science University, Portland, OR) were heterologously expressed in *Escherichia coli* Top10F'. Cells were grown in LB-Miller medium supplemented with 0.9% glycerol, 50 mM MOPS, and 10 mM MgCl$_2$, and expression was induced by adding 0.1% L-arabinose at OD$_{600}$ of 2.5. Proteins were purified as described previously (Ewers *et al*, 2013). Bound ligands were removed using disposable salt exchange columns equilibrated in 20 mM Tris (pH 7.4) containing 500 mM choline-Cl, 1 mM n-dodecyl-β-d-maltoside (DDM).

### Proteoliposome preparation and fluorescence spectroscopy

*Escherichia coli* polar lipid extract (Avanti Polar Lipids, Alabaster, Alabama) and L-α-phosphatidylcholine (Egg, Chicken; Avanti) at a weight ratio of 3:1 were mixed, dried under nitrogen, and dissolved in 100 mM KCl, 20 mM HEPES pH 7.5 (20 mg/ml). The suspension was snap-frozen in liquid nitrogen and then continuously stirred on ice until the suspension was completely thawed. Liposomes were formed by extrusion through 400-nm membranes (Avanti Polar Lipids), diluted with buffer to 4 mg/ml and then stepwise destabilized by several additions of 10% Triton X-100 solution until the OD$_{540}$ value reached 2/3 of the OD$_{540}$ value of the untreated vesicle suspension. Glt$_{Ph}$ was added at a 1:15 protein to lipid weight ratio (LPR 15). The protein/lipid mixture was incubated with gentle agitation at room temperature for 30–45 min before detergent was removed using Bio-Beads SM-2 (Bio-Rad Laboratories, Hercules, California). Bio-Beads were removed by centrifugation (1 min, $1,000 \times g$, 4°C). The supernatant was ultracentrifuged (1 h, $100,000 \times g$, 4°C), the proteoliposome pellets flushed with nitrogen stream, snap-frozen in liquid nitrogen, and stored at −80°C. Successful reconstitution was checked by SDS–PAGE.

For fluorescence measurements, proteoliposomes were thawed and mixed with 20 mM Tris, pH 7.5 and either (i) 100 mM choline-Cl, (ii) 100 mM KCl, or (iii) 100 mM NaCl and 125 μM TBOA. After four freeze–thaw cycles, the proteoliposomes were subjected to bath sonication for 15 min and then to tip sonication for 5 s at 50% intensity, with intermittent pulses and pauses of 200 ms. Tryptophan fluorescence was measured in a Fluorolog spectrofluorometer (Horiba Jobin Yvon, Unterhaching, Germany) in front-face configuration, with excitation at 295 nm and emission at 320 nm. Monochromator slits were set to a bandpass of 5 nm. A sample of 753 μg proteoliposomes was diluted in 2,400 μl buffer in a 1-cm cuvette. Next, 16-doxyl-stearate (16-SASL) was added from a 25 mM stock solution in fluorescence grade ethanol. During titrations, the solution was stirred continuously and fluorescence was monitored. To compare unquenched fluorescence with fluorescence at a single quencher concentration, samples with a higher quencher concentration were incubated for 15 min in the dark after the addition of 60 μM 16-SASL. To obtain A233W Glt$_{Ph}$ fluorescence intensities, photomultiplier counts were corrected for instrument wavelength dependency and excitation intensity. Data were background subtracted and corrected for the error resulting from dilution of the fluorophore. Three independent protein preparations were used to generate the data. CuPh-mediated cross-linking was performed as previously described (Reyes *et al*, 2009).

### Radioactive uptake

Proteoliposomes (0.1 mg Glt$_{Ph}$/ml) were loaded with 200 mM KCl, 20 mM Hepes/KOH (pH 7.4) by three freeze/thaw cycles and extrusion through 400-nm pore size polycarbonate filters (Avestin). The uptake reaction was performed in an Eppendorf Thermomixer and initiated by the addition of 480 μl uptake buffer (200 mM NaCl, 20 mM Hepes/NaOH pH 7.4, 1 μM valinomycin, 156 nM 3H-D-aspartate) to 20 μl proteoliposomes. Uptake buffer and proteoliposomes were pre-equilibrated at 30°C; after the addition of uptake buffer, the mixture was briefly mixed. At a given time point, the reactions were pipetted on pre-washed filters (0.22-μm pore size; GSWP; Millipore) and stopped by immediate suction under vacuum. The filters were washed with 2.5 ml 200 mM LiCl, 20 mM Hepes/NaOH (pH 7.4), mixed with 10 ml Filter-Count (PerkinElmer), and assayed for radioactivity using a TRI-CARB 3110 TR scintillation counter (PerkinElmer). Protein was quantified using an amido black assay.

### Microscale thermophoresis

We performed microscale thermophoresis (MST) experiments (Jerabek-Willemsen *et al*, 2014) to detect $K^+$ binding to Glt$_{Ph}$. Purified and solubilized protein was labeled using the fluorescent reactive dye NT-495-NHS (NanoTemper Technologies GmbH, München, Germany) in 130 mM NaHCO$_3$, 50 mM NaCl, 0.8 mM DDM according to the manufacturer's instructions. Labeled proteins were separated from free un-reacted dye by Ni-NTA-affinity chromatography. For titration experiments, the different $K^+$ concentrations were obtained by serial dilutions (1:2 or 2:3) of the highest ligand concentration. As stock solutions were used (in mM): 3,000 choline chloride, 20 HEPES (NaOH), 0.8 DDM, pH 7.5; 3,000 NaCl, 20 HEPES (NaOH), 0.8 mM DDM, 3,000 KCl, 20 HEPES (KOH), 0.8 mM DDM, pH 7.5. Each buffer was exposed to a temperature jump and the change in fluorescence in the heated volume was recorded over time using a Monolith NT.115 (NanoTemper Technologies GmbH) instrument set to 60% laser power. The steady-state fluorescence after the temperature jump was plotted vs. the ligand concentration to obtain binding curves. Upon the temperature jump, the fluorescence rapidly decreases, which is a property of the dye and does not report on thermophoresis. Therefore, the steady-state fluorescence was normalized to the value measured 0.4 s after the temperature jump (Jerabek-Willemsen *et al*, 2014). The lack of saturation in the MST binding curves renders parameters obtained from fitted Hill equations unreliable. We therefore used non-parametric fits (Nadaraya, 1964; Watson, 1964) in combination with bootstrap sampling for statistical analysis. Each independent binding curve was normalized to the lowest concentration and treated as independent sample for the bootstrap sampling. We then performed non-parametric fits for each bootstrap sample. Parameters for the fitting were local-constant regression with a Gaussian kernel and automatic bandwidth calculation as implemented in pyqt-fit. Mean and 99% confidence intervals from 10,000 bootstrap samples are shown in Fig 4. Significant differences are indicated by non-overlapping confidence intervals.

## Statistical analysis

Simulation data were analyzed with a combination of GROMACS tools (Abraham *et al*, 2015) and in-house python scripts; alchemical free-energy calculations were analyzed with *pmx* (Gapsys *et al*, 2015). Experimental data were analyzed with a combination of FitMaster (HEKA) and python scripts. Current–voltage relationships were generated by plotting the average current amplitudes at the end of a voltage jump against the applied voltage. Current amplitudes were used without any subtraction procedure. Data are given as mean values, with errors determined as standard deviation from independent experiments or obtained via bootstrap sampling, with either complete monomeric trajectories (kinetic parameters) or complete trimeric trajectories at a fixed charge imbalance (charge displacement calculations) treated as independent samples. All statistical evaluations are based on either the two-tailed, unpaired, Student's *t*-test (experimental data), or bootstrapped confidence intervals (MST fit lines; Fig 4, MD data; Figs 6D and 7D).

## Data availability

All relevant data are available from the authors upon request. Modeling datasets and scripts are available at https://github.com/dkortzak/EAAT-K-coupling.

**Expanded View** for this article is available online.

## Acknowledgements
These studies were supported by the Deutsche Forschungsgemeinschaft (DFG, German Research Foundation) to Ch.F. (FA 301/12-1) and J.-P.M. (MA 7525/1-1) as part of the Research Unit FOR 2518, *DynIon*. The authors gratefully acknowledge the computing time granted by the JARA Vergabegremium and provided on the JARA Partition part of the supercomputer JURECA at Forschungszentrum Jülich (Krause and Thörnig, 2018) and the supercomputer CLAIX at RWTH Aachen University. The authors declare no competing financial interests.

## Author contributions
J-PM and CF conceived and supervised the project; DK performed electrophysiological experiments; DK, CA, and J-PM conducted and analyzed MD simulations; AF generated mutant DNA constructs; IW and MIZ produced Glt$_{Ph}$ protein and proteoliposomes; IW performed MST and radiotracer flux experiments; DE performed and analyzed fluorescence spectroscopy experiments; DK and CA prepared the figures; DK, J-PM, and CF wrote the article with comments from all authors.

## Conflict of interest
The authors declare that they have no conflict of interest.

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
