## [Review Process File · The EMBO Journal]

Allosteric gate modulation confers K^+ coupling in glutamate transporters

Daniel Kortzak, Claudia Alleva, Ingo Weyand, David Ewers, Meike I. Zimmermann, Arne Franzen, Jan-Philipp Machtens and Christoph Fahlke

Review timeline:	Submission date:	3rd Jan 2019
	Editorial Decision:	27th Feb 2019
	Editorial Correspondence:	1st Mar 2019
	Revision received:	24th May 2019
	Editorial Decision:	12th Jun 2019
	Revision received:	30th Jul 2019
	Accepted:	5th Aug 2019

Editor: Daniel Klimmeck

Transaction Report:

1st Editorial Decision

27th Feb 2019

Thank you for the submission of your manuscript (EMBOJ-2019-101468) to The EMBO Journal. Please again accept my apologies for the unusual delay in the peer-review process at this time of the year. Your manuscript has been sent to three referees, but we have received reports from only two of them, which I enclose below. The third referee got much delayed and has not sent been able to send us his-her report so far. However, based on the other two overall positive reports and in the interest of time, we have now decided to proceed with our decision. As outlined below, we can invite you to revise your manuscript, pending no major concerns regarding technical robustness of the study will be brought up by referee #3.

As you will see, the referees acknowledge the potential interest and novelty of your work, although they also express a number of issues that will have to be addressed before they can support publication of your manuscript in The EMBO Journal. In more detail, the referees ask you to corroborate your findings by additional controls regarding potassium dialysis and alternative mutants (ref#2 and ref#1, pts. 3,4) and requests a more detailed discussion (ref#1, pts 1, 2). We agree that these points would need to be conclusively addressed to achieve the level of robustness needed for The EMBO Journal.

I judge the comments of the referees to be generally reasonable and as mentioned, given their overall interest, we are in principle happy to invite you to revise your manuscript experimentally to address the referees' comments.

REFeree REPORTS:

Referee #1:

This manuscript addresses an important issue on the mechanism of glutamate transport, namely the

role of potassium coupling in mammalian glutamate transporters. These transporters mediate co-transport of glutamate or aspartate with 3 sodium ions and one proton and counter-transport with one potassium ion. Archeal aspartate transporters have a simpler stoichiometry and operate by co-transport of aspartate with 3 Na ions; without the requirement of proton co-transport and potassium counter-transport. Structural studies on the archeal transporters have identified the three sodium binding sites Na1, Na2 and Na3 and in the only mammalian structure available the same Na2 site was visualized. Occupation of this site is required for the closure of the so-called HP2 gate, which is required for translocation of the substrate-bound transport domain by the well-known elevator movement. Based on conservation of the amino acid residues participating in the two other sites of the mammalian transporters are most probably identical. The probable proton binding site has also been inferred. On the other hand, the location of the potassium site is not known. Moreover, another important question is why the potassium site is needed in the mammalian transporters but not in their archeal counterparts. It is exactly these two questions which the Authors of this manuscript are trying to answer.

Because a structure of a mammalian transporter with a bound potassium ion is not available, the questions have been addressed using extensive molecular dynamics simulations on the mammalian EAAT1 transporter, followed by experimentally testing several of the ensuing predictions. Four potential potassium sites are identified but only one, the K1 site, meets the criteria for a physiological relevant site. Simulations indicate that the potassium does not bind directly to K1 but does this after transfer of the potassium from one of the other sites (mostly K2). The binding to K1 causes, via an allosteric mechanism, the closure of the HP2 gate. This closure is required for the movement of the transport domain resulting in translocation of the substrate-free transporter (the so-called return step). It was shown that the archeal transporters can also bind potassium, but the propensity for gate closure is much higher here, even in the absence of potassium. An Alanine residue at position 360 (448 in EAAT1) of GltPh is a major determinant for the potassium independent gate closure, but in the mammalian transporters a leucine residue occupies this position. Remarkably, the mutation of a leucine residue in the HP2 gate of the mammalian transporter to alanine, results in potassium-independent transport currents. Moreover, evidence is presented that the closing of this loop in the mammalian transporters is counteracted by a conserved ion pair. This manuscript is important and well-designed, but there are several issues which need to be addressed.

Critique

Major

1. The location of the potassium site, formed by amino acid residues from Na1 and Na3 sites is in nice agreement with extensive data from structure-function studies on the mammalian transporters. What is missing here is a discussion on if or why the absence of a requirement for potassium also leads to that for the proton. The proton acceptor of the mammalian transporters, a glutamate at position 318 (according to the numbering of the archeal GltPh transporter; see my minor comment) is replaced by a glutamine in GltPh. A question is if and how this relates to the absence of the potassium translocation. Does GltPh-Q318E mediate proton dependent, potassium independent transport?
2. Fig. 4: The lack of glutamate induced currents in some of the mutants is due to the fact that the side-chains of the mutated residues participate in the substrate binding site. This should be mentioned for the benefit of the readers.
3. Fig. 6: Glutamate-induced currents are dependent on potassium in the mammalian transporter EAAT1, but not in the L448A mutant. Does the dialysis remove all the internal potassium? This should be dealt with by measuring radioactive transport upon reconstitution in liposomes. This measurement is also more sensitive than that of the transport currents and the Authors know how to do that (see Fig. 5D). It is important to show an I/V plot of the transport currents.
4. Fig. 7: Serine-induced currents are shown for R397A (the Arg is a partner of the ion pair). This is unexpected and in contrast to the role of this Arg, where its replacement to a neutral amino acid leads to exchange and not net flux. This is backed up by structural information of the substrate-free GltTk transporter. What do they find for E318 (the ion pair partner of R397) mutants? These mutants have been shown to be obligate exchangers.

Minor

1. Please use the same numbering for GltPh and EAAT1 (and mention this of course in the beginning). Otherwise it will be difficult for the readers to keep track of what is going on.
2. Does the D312N mutant lack the potassium dependence of the MST like D404N?

Referee #2:

General Comments

The article by Kortzak provides a very provocative proposal for the mechanism for K⁺ coupling of Glutamate Transporters. This has been a topic of considerable interest for many years because it would provide the missing piece in understanding the overall mechanism by which Na⁺, H⁺ and K⁺ ions are coupled to the transport of glutamate across the membrane. Other groups have attempted to address the question of K⁺ coupling with varying degrees of success. Whilst none of the previous studies can be classified as being definitive, the most compelling study was done by Verdon et al. (2014) *eLife* 3, e02283 and they used crystallography to suggest that K⁺ bound to a site that overlaps with the aspartate/glutamate binding site. This study was based on thalium binding, but the coordination of the thalium ion was consistent with it representing a K⁺ binding site, and it was not possible to displace the thalium with potassium and so the conclusions remained tentative. A previous molecular dynamics study by Holley and Kavanaugh (2009) made similar predictions as to the location of the K⁺ site. Furthermore, the related prokaryotic aspartate transporter GltPh is not coupled to K⁺ countertransport, yet all the proposed K⁺ binding residues are conserved between the EAATs and GltPh. With this background, Kortzak et al have used unguided MD simulations using the crystal structure of EAAT1 as a starting point for the simulations to identify possible K⁺ binding sites. 4 potential binding sites have been identified in EAAT1 and 3 of these sites are also found in GltPh, suggesting that the nature of the differences between the EAATs and GltPh will be subtle. A primary K⁺ site has been identified termed K1. They also predicted that K⁺ ions first bind to K2/K3 sites before reaching the K1 site. The K1 site is different to the thalium binding site identified by Verdon et al (2009), but the K3 site is similar, but not identical to the site predicted by Verdon/Holley. The role of the K1 site is further investigated to explain the subtle differences between the EAATs and GltPh.

Minor concern

The role of the stabilising of HP2 as an explanation for subtle differences is very novel, very interesting and the data is consistent with interpretation. This idea provides a compelling argument for the idea of the way that HP2 closes has an impact on K⁺ coupling, but I think it should be based on more than one conservative mutation. The L448A data is very nice, but additional mutations of other residues in HP2 showing similar effects would make the concept more convincing rather than relying on just one mutation. This could be done through simulations using alternate amino acids at this site, or mutations of neighbouring residues.

Overall, this is a well constructed study addressing an important topic that has been difficult to come to terms with. The proposal put forward in this study is convincing, but I dare say it will not be the end of the story. One minor limitation, that should probably be pointed out in the discussion, is the use of the EAAT1 (crystal structure) for MD studies rather than the WT EAAT1 structure. The crystal structure contains a large number of mutations, some of these mutations could lead to subtly altered structural predictions compared to the WT structure and given the subtle differences that lead to K⁺ coupling some of these differences may generate spurious conformations.

Editorial Correspondence

1st Mar 2019

Please find enclosed the delayed comments of referee #3 on your manuscript EMBOJ-2019-101468 for your consideration.

This referee expresses concerns regarding the integration of earlier literature and structures, data interpretation as well as additional mutant experiments and controls, which in his-her view would be required to consolidate your findings and model.

Please consider these comments in your revision and let me know if you need additional input from our side or the referees on the matter.

Referee #3

The manuscript by the Fahlke group presents a combination of simulation studies and experimental work to describe potassium ion binding to the prokaryotic transporter GltPh and to mammalian transporters EAAT1 and 2.

I cannot judge the quality of the MD simulations, but I have a few major concerns about the interpretations of the experimental data, as well as the author's interpretation of data from existing literature, especially on the GltPh transporters.

Major points:

The authors appear to cite incorrectly what was published on HP2 opening in the IFC. For instance on page 22: "HP2 opening in inward facing gltph (reyes et al) was demonstrated by TBOA binding". The cited manuscript does not show HP2 opening, just TBA (not TBOA) binding. The manuscript also clearly states that the gating in the IFC cannot be explained by HP2 movement alone. The citation is therefore misplaced. Similarly on page 13: the citation of Verdon is out of context. The manuscript of Verdon shows subtle conformational changes of HP1, but no changes in conformation of HP2. It is incomprehensible how the authors can then conclude that the work of Verdon is consistent with the notion that HP2 acts as a gate in the IFC. To be short: There is no experimental evidence that HP2 is a gate of the IFC. The only indication comes from simulations, which may be correct, but at the moment not supported by experiments.

The experimental evidence provided by the authors that GltPh binds potassium ions is weak. The MST data show a marginal difference between the mutant and the wt. This is not convincing, especially, because the mutant is inactive in transport, and may not be able to sample conformations that allow for potassium binding. A mutant should be used that is still active in transport.

Also the experiment shown in figure 5F is not convincing. Crucial control experiments (for instance by adding both Na and K in the TBOA experiment; Na only addition; use of a mutant fixed by crosslinking in combination with the different ions) are needed to make a convincing case.

The authors appear to have overlooked a crucial consequence of their model. If GltPh in some conditions may counter transport K (even if it is not essential, and may happen only in a fraction of transport cycles), this will affect the reversal potentials. The Mindell group has recently published a method to do such measurements with prokaryotic transporters. The authors should do these experiments at different internal K concentrations. If the reversal potentials are dependent on the internal K concentration (e.g. between 1 and 300 mM, concentrations still sufficient for clamping of the voltage using valinomycin), it would be a strong argument in favor of the K binding hypothesis. If there is no different, K binding would not be relevant under physiological conditions.

On page 15 the authors describe conformational changes in the NMDGT motif upon K binding. There are structures of homologues (Jensen 2013 NSMB, Guskov 2016 Nature Comm) in the apo state, yet in the presence of K. Are the changes in conformation of the NMDGT motif observed in these structures? Do the authors think these apo structures are in fact K-bound. Discussion needed.

Page 4: "Apo states were modelled by removing bound Na or Asp molecules from the structures". I do not understand why the authors have not used the published apo structures (Verdon, 2014, Jensen, 2013, Guskov, 2016

1st Revision - authors' response

24th May 2019

Please see next page.

We would like to submit a revised version of our manuscript entitled *Principles of K⁺ coupling in the glutamate transporter family* by Daniel Kortzak, Claudia Alleva, Ingo Weyand, David Ewers, Meike I. Zimmermann, Arne Franzen, Jan-Philipp Machtens and Christoph Fahlke to be considered for publication in *The EMBO Journal*. The referees provided clear and constructive criticisms that we have fully addressed in the following text and in the revised manuscript. Changes in the manuscript are marked as green, and specific responses to the referees' comments and other manuscript changes are outlined below.

Reviewers' Comments:

Reviewer #1: *This manuscript addresses an important issue on the mechanism of glutamate transport, namely the role of potassium coupling in mammalian glutamate transporters. These transporters mediate co-transport of glutamate or aspartate with 3 sodium ions and one proton and counter-transport with one potassium ion. Archeal aspartate transporters have a simpler stoichiometry and operate by co-transport of aspartate with 3 Na ions; without the requirement of proton co-transport and potassium counter-transport. Structural studies on the archeal transporters have identified the three sodium binding sites Na1, Na2 and Na3 and in the only mammalian structure available the same Na2 site was visualized. Occupation of this site is required for the closure of the so-called HP2 gate, which is required for translocation of the substrate-bound transport domain by the well-known elevator movement. Based on conservation of the amino acid residues participating in the two other sites of the mammalian transporters are most probably identical. The probable proton binding site has also been inferred. On the other hand, the location of the potassium site is not known. Moreover, another important question is why the potassium site is needed in the mammalian transporters but not in their archeal counterparts. It is exactly these two questions which the Authors of this manuscript are trying to answer.*

Because a structure of a mammalian transporter with a bound potassium ion is not available, the questions have been addressed using extensive molecular dynamics simulations on the mammalian EAAT1 transporter, followed by experimentally testing several of the ensuing predictions. Four potential potassium sites are identified but only one, the K1 site, meets the criteria for a physiological relevant site. Simulations indicate that the potassium does not bind directly to K1 but does this after transfer of the potassium from one of the other sites (mostly K2). The binding to K1 causes, via an allosteric mechanism, the closure of the HP2 gate. This closure is required for the movement of the transport domain resulting in translocation of the substrate-free transporter (the so-called return step). It was shown that the archeal transporters can also bind potassium, but the propensity for gate closure is much higher here, even in the absence of potassium. An Alanine residue at position 360 (448 in EAAT1) of GltPh is a major

determinant for the potassium independent gate closure, but in the mammalian transporters a leucine residue occupies this position. Remarkably, the mutation of a leucine residue in the HP2 gate of the mammalian transporter to alanine, results in potassium-independent transport currents. Moreover, evidence is presented that the closing of this loop in the mammalian transporters is counteracted by a conserved ion pair. This manuscript is important and well-designed, but there are several issues which need to be addressed.

Critique

Major

1. The location of the potassium site, formed by amino acid residues from Na1 and Na3 sites is in nice agreement with extensive data from structure-function studies on the mammalian transporters. What is missing here is a discussion on if or why the absence of a requirement for potassium also leads to that for the proton. The proton acceptor of the mammalian transporters, a glutamate at position 318 (according to the numbering of the archeal GltPh transporter; see my minor comment) is replaced by a glutamine in GltPh. A question is if and how this relates to the absence of the potassium translocation. Does GltPh-Q318E mediate proton dependent, potassium independent transport?

We agree with reviewer 1 that this is a very interesting question. As suggested we reconstituted, purified and solubilized Q318E Glt_{Ph} in liposomes and studied radioactive ³H-aspartate uptake for two pH values. Aspartate uptake turned out to be proton independent for WT and mutant transporters (Fig. 1 for reviewer).

Figure 1 for reviewer. Q318E does not couple Glt_{Ph} aspartate transport to proton gradients. Initial rates of WT or Q318E Glt_{Ph} ³H-aspartate uptake for two external pH (mean ± SD, 4 experiments). Proteoliposomes were loaded with 100mM KCl, 20mM PIPES, pH 7.5 and diluted 133-fold into reaction buffer containing 100mM NaCl, 1µM valinomycin 1µM l-[³H]aspartate, 20mM PIPES, pH 7.5 or 6.0.

We furthermore tested whether the partially K⁺-uncoupled mutant L447A EAAT2 is still pH dependent. In whole-cell patch-clamp experiments with cells expressing L447A EAAT2, glutamate-uptake currents are similarly modified by changing the external pH from 7.4 to 8.5 as WT EAAT2. This finding demonstrates that EAATs can simultaneously be H⁺ dependent and K⁺ uncoupled. We now show the results in Fig. S9 and discuss this issue shortly on p.10, line 27.

Understanding potential relationships between K⁺ and H⁺ coupling will be an important step towards a complete mechanistic understanding of EAAT transport mechanisms. However, we believe addressing this open question first awaits *in-depth* analysis of the H⁺ coupling

mechanism and will require additional experiments and simulations that are out of the scope of the current manuscript.

2. *Fig. 4: The lack of glutamate induced currents in some of the mutants is due to the fact that the side-chains of the mutated residues participate in the substrate binding site. This should be mentioned for the benefit of the readers.*

Indeed, this is true for one of the studied mutants, N401A Glt_{ph}/N482A EAAT2. However, N482A leaves K⁺ coupling in EAAT2 intact and thus serves as an important control for the conclusion of Fig. 4. We added this information on p. 7, line 11.

3. *Fig. 6: Glutamate-induced currents are dependent on potassium in the mammalian transporter EAAT1, but not in the L448A mutant. Does the dialysis remove all the internal potassium? This should be dealt with by measuring radioactive transport upon reconstitution in liposomes. This measurement is also more sensitive than that of the transport currents and the Authors know how to do that (see Fig. 5D).*

Pusch and Neher (*Pflügers Arch* 411, 204-211 (1988)) performed a comprehensive analysis of the control of intracellular ion concentrations via dialysis through the patch pipette. They observed a K⁺ diffusion rate that was higher as for all other tested components of the pipette solutions, indicating that intracellular [K⁺] can be very well controlled through patch pipette-mediated dialysis. [K⁺] exchange time constants depend on the cell size and on the series resistance, and we employed the quantitative analysis provided in this paper to predict exchange time constants in our experiments. Using mean series resistances ($5.2 \pm 1.9 \text{ M}\Omega$) and cell capacitances ($20.7 \pm 7.5 \text{ pF}$) from our experiments, exchange of pipette [K⁺] with the cytoplasm occurs with a time constant of around 15 s, so that intracellular [K⁺] falls below 1 μM within approximately 75 s when using K-free pipette solution. We regularly waited at least 3 min after opening the cell before starting the experiments. This information was added to the Material and Methods section on p.21, line 6.

We neither observed transport currents in cells dialyzed with K⁺-free solutions for WT nor for 12 out of 16 EAAT1/EAAT2 mutants, which we engineered based on sequence variations between EAATs/Glt_{ph}. Only four of the 16 tested mutants (L447A EAAT2, L448A EAAT1, L448T EAAT1, and R478A EAAT2, some of which were tested as additional mutations during the revision in response to reviewer 2) mediated K⁺-independent transport currents upon dialysis with K⁺-free solutions. We carefully analyzed our experimental conditions by testing whether series resistances or cell capacitances were especially high in recordings interpreted as K⁺-independent uptake currents and could exclude such a correlation (Fig. 2 for reviewer). At comparable series resistances cells expressing L448A EAAT1/L447A EAAT2 exhibited glutamate uptake currents, whereas cells expressing WT EAAT1/EAAT2 did not. Thus, there remains little doubt that K⁺-free pipette solutions indeed resulted in very low [K⁺]_{int} in our experiments.

Figure 2 for reviewer. Amplitudes of glutamate transport currents in cells dialyzed with K^+ -free solutions do not depend on the series resistance or cell capacitance. Violin plots of glutamate transport currents at -140mV versus series resistance or cell capacitance for whole-cell patch-clamp experiments with WT EAAT1/EAAT2 or L448A EAAT1/L447A EAAT2.

However, our experiments do not exclude the—albeit unlikely—modification of K^+ -binding sites towards extremely high affinity. In this case, even a very low residual intracellular $[K^+]_i$ could support obligatory K^+ -dependent uptake, and even radioactive transport after reconstitution in liposomes—which permits more direct control of internal $[K^+]_i$ than whole-cell dialysis—might not allow refuting this scenario. We therefore refrained from such experiments and rather decided to perform additional experiments, in which changes in transport stoichiometry can be tested without relying on a complete removal of internal K^+ . Using an approach pioneered by Zerangue and Kavanaugh (*Nature* 383, 634-637 (1996)) we measured transport current reversal potentials for WT and L447A EAAT2 upon changing $[K^+]_o$ from 100 mM, to 50 mM and 25 mM. We chose EAAT2 in these experiments because of the higher uptake current amplitudes of this isoform.

In these experiments, glutamate, Na^+ , and K^+ were present on the intra- and on the extracellular membrane sides, and the extracellular $[K^+]_o$ was modified. Permeant anions were substituted with gluconate to abolish EAAT-associated anion currents, and transport currents were measured as TBOA-sensitive current components. We quantified shifts of transport current reversal potential and compared it with calculated transport current reversal potentials, at which the transporter's driving force equals zero (new Fig. S9). Experiments with cells expressing WT EAAT2 provided reversal potentials that were in full agreement with the predictions for a coupling stoichiometry of 3 Na^+ , 1 H^+ , and 1 glutamate in exchange with 1 K^+ . In contrast, for L447A EAAT2, we observed changes in the reversal potential upon changed $[K^+]_o$ that were less pronounced. These findings indicate that L447A EAAT2 can mediate glutamate transport both in exchange with one K^+ or without K^+ exchange. We now describe these novel experiments on p.10, line 15 and in Fig. S9.

Taken together, the two experiments, measurements of transport currents upon dialysis with K^+ -free solutions and determination of transport current reversal potentials, confirm that K^+ -independent transporters can be generated by modifying the HP2 dynamics.

It is important to show an I/V plot of the transport currents.

As requested we now show I–V plots of transport currents in Figs. 6, 7, and S9.

4. Fig. 7: Serine-induced currents are shown for R397A (the Arg is a partner of the ion pair). This is unexpected and in contrast to the role of this Arg, where its replacement to a neutral amino acid leads to exchange and not net flux. This is backed up by structural information of the substrate-free GltTk transporter. What do they find for E318 (the ion pair partner of R397) mutants? These mutants have been shown to be obligate exchangers.

Voltage-clamp recordings and uptake experiments on oocytes expressing R447C EAAT3 (Glt_{ph} R397) revealed electroneutral cysteine uptake, suggesting that this mutation either changes the Na^+ coupling stoichiometry or converts EAAT3 into a homoexchanger (Bendahan *et al. J Biol Chem* 275, 37436-37442 (2000)). Recent spectroscopic experiments supported the second theory by demonstrating that the reduced apparent aspartate affinity after substitution of the equivalent side chain R397 in Glt_{ph} is due to impaired coupling between aspartate binding and $[Na^+]$ (Focke *et al. Biochemistry* 54, 1694-1702 (2015)).

EAAT1 simulations demonstrated reversible salt-bridge interactions between the homologous R479 and E406 that stabilize open HP2 and prevent translocation in the apo state unless K^+ binds. To challenge this observation experimentally, we chose the R479A substitution to prevent any polar or charged interaction, for which we reproducibly measure robust serine uptake currents in presence in absence of K^+ (Fig. 7). We conclude that different arginine substitutions differ in their impact on coupled transport activity, but do not necessarily impair coupled transport.

We expressed E406D and E406Q EAAT1 in mammalian cells and studied uptake and anion currents via patch-clamp recordings. We observed robust anion currents for both mutants that are activated under exchange conditions. However, we could not detect any transport currents and conclude that both mutants are exchangers (Fig 3 for reviewer).

Figure 3 for reviewer. E406D/E406Q EAAT1 function as obligate exchangers. A,B,C, Voltage dependence of steady-state current amplitudes after dialysis with NO_3^- (A,B) or gluconate-based (C) internal solutions. E406D/E406Q EAAT1 anion currents were measured in Na^+ or K^+ -based solutions with and without glutamate (A,B). Transport currents were calculated as glutamate-sensitive current components in a Na gluconate-based external solution (C) (mean \pm SD, 4 experiments).

Minor

1. Please use the same numbering for GltPh and EAAT1 (and mention this of course in the beginning). Otherwise it will be difficult for the readers to keep track of what is going on.

We now provide for every amino acid exchange in EAAT1 and EAAT2 the homologous residue in Glt_{Ph} in parenthesis. The information is given on p.4, line 15.

2. Does the D312N mutant lack the potassium dependence of the MST like D404N?

We performed MST experiments on D312N Glt_{Ph} and added the results to Fig. 4. Similar to D405N, D312N significantly reduces K⁺-dependent changes in thermophoretic mobility of Glt_{Ph}. Using non-parametric regression and bootstrap sampling, we demonstrate that both mutations cause a statistically significant shift in the K⁺ dependence of the MST signal.

D312N is expected to impair Glt_{Ph} transport activity due to altered Na⁺ interactions (Bastug *et al. PloS One* 7:e33058 (2012)). However, mutant EAAT3 carrying the same amino acid exchange retains the ability to bind substrates and undergo substrate-induced conformational changes (Tao *et al. J Biol Chem* 281, 10263-10272 (2006)) (p.7, line 19).

Reviewer #2: *The article by Kortzak provides a very provocative proposal for the mechanism for K⁺ coupling of Glutamate Transporters. This has been a topic of considerable interest for many years because it would provide the missing piece in understanding the overall mechanism by which Na⁺, H⁺ and K⁺ ions are coupled to the transport of glutamate across the membrane. Other groups have attempted to address the question of K⁺ coupling with varying degrees of success. Whilst none of the previous studies can be classified as being definitive, the most compelling study was done by Verdon *et al.*, (2014) *eLife* 3, e02283 and they used crystallography to suggest that K⁺ bound to a site that overlaps with the aspartate/glutamate binding site. This study was based on thalium binding, but the coordination of the thalium ion was consistent with it representing a K⁺ binding site, and it was not possible to displace the thalium with potassium and so the conclusions remained tentative. A previous molecular dynamics study by Holley and Kavanaugh (2009) made similar predictions as to the location of the K⁺ site. Furthermore, the related prokaryotic aspartate transporter GltPh is not coupled to K⁺ countertransport, yet all the proposed K⁺ binding residues are conserved between the EAATs and GltPh. With this background, Kortzak *et al* have used unguided MD simulations using the crystal structure of EAAT1 as a starting point for the simulations to identify possible K⁺ binding sites. 4 potential binding sites have been identified in EAAT1 and 3 of these sites are also found in GltPh, suggesting that the nature of the differences between the EAATs and GltPh will be subtle. A primary K⁺ site has been identified termed K1. They also predicted that K⁺ ions first bind to K2/K3 sites before reaching the K1 site. The K1 site is different to the thalium binding site identified by Verdon *et al* (2009), but the K3 site is similar, but not identical to the site predicted by Verdon/Holley. The role of the K1 site is further investigated to explain the subtle differences between the EAATs and GltPh.*

Minor concern

The role of the stabilising of HP2 as an explanation for subtle differences is very novel, very interesting and the data is consistent with interpretation. This idea provides a compelling argument for the idea of the way that HP2 closes has an impact on K⁺ coupling, but I think it should be based on more than one conservative mutation. The L448A data is very nice, but additional mutations of other residues in HP2 showing similar effects would make the concept more convincing rather than relying on just one mutation. This could be done through simulations using alternate amino acids at this site, or mutations of neighbouring residues.

We tested 12 amino acid exchanges between EAAT1 and Glt_{Ph} (Supplemental Table 6) for the original version of the manuscript and identified two point-mutations (L448A and R479A EAAT1) that rendered glutamate transport K⁺ independent in our experiments, consistent with our simulations (Fig. 6 and 7). The low number of effective mutations is not surprising in light of the extensive mutagenesis studies that have been performed by many groups worldwide in search for sequence determinant of K⁺ coupling during the last 30 years.

In response to this comment of reviewer 2, we now added experimental data on two more mutations in L448 (L448T and L448V EAAT1) (Fig. 6 and S9) and quantified one additional mutation (L448T) in MD simulations (Fig. 6). L448V EAAT1 is fully K⁺-dependent, whereas L448T EAAT1 exhibits transport currents also in the absence of K⁺. This behavior is in full agreement with novel simulation data on L448T EAAT1.

We also tested glutamate transport by L448G EAAT1, but obtained inconclusive results on this mutant. In cells expressing L448G EAAT1, glutamate application in the absence of internal K⁺ results in the enhancement of currents in both directions, with a reversal potential of 0 mV, whereas glutamate induced currents do not reverse with K⁺ in the pipette (Fig 4 for reviewer). Obviously, L448G EAAT1 mediates transport processes that are not existent in WT EAAT1. The glycine insertion might disrupt the α -helical backbone structure of HP2 and may thus induce additional unknown changes in HP2 structure and dynamics, which prevent meaningful interpretation of these experiments and which cannot be reliably studied using MD simulations. We therefore refrain from any mechanistic interpretation of L448G EAAT1. Since these results do not add to understanding K⁺ coupling in the EAATs, we decided not to show them in the current paper. However, we will certainly study this novel behavior in the future.

Figure 4 for reviewer. L448G EAAT1 exhibits a novel current component in K⁺-free solutions. Plot of glutamate-sensitive L448A or L448G EAAT1 currents from cells intracellularly dialyzed with choline-gluconate or Kgluconate-based solutions (mean \pm SD, 4 experiments).

Overall, this is a well constructed study addressing an important topic that has been difficult to come to terms with. The proposal put forward in this study is convincing, but I dare say it will not be the end of the story. One minor limitation, that should probably be pointed out in the discussion, is the use of the EAAT1 (crystal structure) for MD studies rather than the WT EAAT1 structure. The crystal structure contains a large number of mutations, some of these mutations could lead to subtly altered structural predictions compared to the WT structure and given the subtle differences that lead to K⁺ coupling some of these differences may generate spurious conformations.

The EAAT1 X-ray structure includes thermostabilizing mutations, which were necessary for expression and crystallization (Canul-Tec *et al. Nature* 544, 446-451 (2017)). None of the thermostabilizing EAAT1 mutations affect residues involved in ion or substrate binding, and the authors demonstrated robust Na⁺- and K⁺-dependent glutamate uptake activity for mutant transporters in radioactive uptake experiments. The overall protein fold of the outward-facing EAAT1 structure is very similar to Glt_{ph} crystal structures (Boudker *et al. Nature* 445, 387-393(2007)), and the structure of the superimposed EAAT1 transport domain is also very similar (with less than 1 Å RMSD) to the inward-facing, wild-type, human ASCT2 structure obtained by cryo-electron microscopy (Garaeva *et al. Nat Struct Mol Biol* 25, 515-521 (2018)). We are therefore convinced that the used EAAT1 X-ray structure represents the most reasonable structural model for EAATs available to date. However, we agree with reviewer 2 that one cannot fully exclude small structural alterations caused by the thermostabilizing mutants and we now address potential limitations caused by using the thermostabilized EAAT1 structure in our simulations on p. 12, line 19.

Reviewer #3. *The manuscript by the Fahlke group presents a combination of simulation studies and experimental work to describe potassium ion binding to the prokaryotic transporter GltPh and to mammalian transporters EAAT1 and 2.*

I cannot judge the quality of the MD simulations, but I have a few major concerns about the interpretations of the experimental data, as well as the author's interpretation of data from existing literature, especially on the GltPh transporters.

Major points:

The authors appear to cite incorrectly what was published on HP2 opening in the IFC. For instance on page 22: "HP2 opening in inward facing gltph (reyes et al) was demonstrated by TBOA binding". The cited manuscript does not show HP2 opening, just TBA (not TBOA) binding. The manuscript also clearly states that the gating in the IFC cannot be explained by HP2 movement alone. The citation is therefore misplaced. Similarly on page 13: the citation of Verdon is out of context. The manuscript of Verdon shows subtle conformational changes of HP1, but no changes in conformation of HP2. It is incomprehensible how the authors can then conclude that the work of Verdon is consistent with the notion that HP2 acts as a gate in the IFC. To be short: There is no experimental evidence that HP2 is a gate of the IFC. The only indication comes from simulations, which may be correct, but at the moment not supported by experiments.

We agree that none of the cited references reports direct experimental proof for HP2 opening in the IFC and improved the discussion of the existing literature to the role of HP2 on p. 13, line 6.

MD simulations by Zomot and Bahar (*J Biol Chem* 28, 8231 - 8237 (2013)) demonstrated that helical hairpin HP2 and not HP1 serves as an intracellular gate, in full agreement with our own simulations (Fig. S7). Published experimental data support that HP2 can open in the IFC.

Outward-facing crystal structures of Glt_{ph} and EAAT1 demonstrate how the benzyl moiety of the competitive agonist TBOA bound to the substrate-binding pocket keeps HP2 open (Boudker *et al. Nature* 445, 387-393 (2007), Canul-Tec *et al. Nature* 544, 446-451 (2017)). Thus far, no inward-facing TBOA-bound structure is available, and HP2 is tightly packed against the trimerization domain in inward-facing Glt_{ph} crystal structures. However, using isothermal titration calorimetry, Reyes *et al* demonstrated that both TBA and TBOA (Suppl. Fig. 6 of the paper) can bind to outward- and inward-facing Glt_{ph}, with similar binding free energies (Reyes *et al. Nat Struct Mol Biol* 20: 634-640 (2013)). Furthermore, Oh and Boudker used stopped-flow fluorescence spectroscopy experiments to identify a conformational selection binding mechanism for TBOA to constrained inward-facing Glt_{ph} (Oh & Boudker *eLife* 7 (doi: 10.7554/eLife.37291, 2018)). Since the substrate-binding pocket does not change conformation upon transition from the outward- to the inward-facing states (Reyes *et al. Nature* 462, 880-885 (2009)), these data provide strong support for the notion that HP2 can open in the inward-facing conformation.

One might imagine that HP2 opening may be accomplished via minor lateral movement of the transport domain away from the trimerization domain: Such an ‘unlocked’, inward-facing conformation that would sterically permit HP2 opening has been determined for a Glt_{ph} mutant with increased transport activity (Akyuz *et al. Nature* 502, 114-118 (2013)).

Concerning the citation of Verdon *et al.* on page 13, we would like to apologize for this misplaced citation, which we corrected in the revised version (p. 13, line 9).

The experimental evidence provided by the authors that GltPh binds potassium ions is weak. The MST data show a marginal difference between the mutant and the wt. This is not convincing, especially, because the mutant is inactive in transport, and may not be able to sample conformations that allow for potassium binding. A mutant should be used that is still active in transport.

K⁺ does cause alteration in WT mobility that are significantly different and right-shifted in D405N or D312N Glt_{ph} (which was added in response to reviewer 1). We now describe the statistical treatment of these data in more detail on p.23, line 18. To facilitate comparison of WT with both mutations we repeated experiments on WT and D405N Glt_{ph} and now show these novel data in Fig 4C.

To test for functionality of D405N Glt_{ph} we performed radioactive uptake experiments that are shown in Fig. S6 and described on p. 7, line 22. D405N does not abolish Glt_{ph} transport. In contrast, D312N Glt_{ph} is assumed to be transport incompetent as shown for D312A Glt_{ph} in Bastug *et al. (PloS one* 7: e33058 (2012)). However, EAAT3 experiments indicate that this mutant still retains the ability to bind substrates and to undergo substrate-induced conformational changes (Tao *et al. J Biol Chem* 281, 10263-10272 (2006)) (p.7, line 19).

MST studies changes in mobility in aqueous solutions and permits quantifying substrate binding that results in measurable changes in thermomobility via conformational changes upon substrate association. In our case, K⁺ may increase Glt_{ph} thermophoretic mobility by stimulating HP2 closure. The changes in Glt_{ph} mobility upon K⁺ addition are comparable to published MST

results. For example, Parker and Newstead (*Nature* 507, 68-72, (2014)) studied the nitrate transporter NRT1.1 from *Arabidopsis thaliana* and observed signals upon application of NO_3^- as we observed for K^+ -binding to Glt_{Ph} . Figure 5 for reviewer depicts an MST aspartate titration experiment on Glt_{Ph} , demonstrating again comparable signals as observed for K^+ .

Figure 5 for reviewer. Aspartate binding causes similar MST signals as K^+ . Normalized changes in fluorescence in MST experiments vs. aspartate concentrations (mean \pm SD, 6 experiments). Fit line and confidence intervals are calculated as described in the manuscript (p.24, line 4).

Also the experiment shown in figure 5F is not convincing. Crucial control experiments (for instance by adding both Na and K in the TBOA experiment; Na only addition; use of a mutant fixed by crosslinking in combination with the different ions) are needed to make a convincing case.

In response to this criticism we studied additional conditions in crosslinked and non-cross-linked transporters and are now showing old and new data in Fig. S6. We did not perform experiments with Na/K mixtures, since we felt that the heterogeneity of protein conformations under these conditions preclude interpretation of the experimental data.

As expected, A233W was significantly more quenched in Glt_{Ph} after cross-linking in the inward-facing conformation. We observed pronounced differences between cross-linked and not cross-linked Glt_{Ph} in Na^+ +TBOA, in good agreement with the notion that Na^+ +TBOA preferentially holds the transporter in an outward-facing conformation, as well as with K^+ as the only substrate. To our initial surprise, we observed more quenching for cross-linked transporters in K^+ than in Na^+ +TBOA. This result is most likely due to incomplete cross-linking (Ewers *et al. Proc Natl Acad Sci U S A* 110: 12486-91 (2013)), with a certain percentage of transporters kept in the outward-facing conformation by bound TBOA.

One control condition, Na^+ only, demonstrated a limitation of the A233W quenching experiment we were not aware before. We observed intermediate fluorescence quenching of A233W Glt_{Ph} under these conditions, likely due to stabilization of both inward-facing and outward-facing conformations. Published data on the Glt_{Ph} conformation are inconclusive regarding the conformational distribution of the transporter in presence of Na^+ only. Erkens *et al. (Nature* 502: 119-123 (2013) and Hänelt *et al. (Nat Struct Mol Biol* 20: 210-214 (2013)) reported different preferences for Glt_{Ph} in Na^+ that depended not only on applied ion gradients and voltages, but also on the position of the inserted reporters. Others suggested that

transporters preferentially assume the OFC under Na⁺ only conditions (Ruan *et al. Proc Natl Acad Sci U S A* 114: 1584-1588 (2017)).

Thus, average A233W quenching in the presence of Na⁺ (resulting in fractions of transporters kept in the inward or in the outward-facing conformations) is comparable with quenching in K⁺ (under these conditions the transporter is more prone to translocate). Although the A233W quenching assay provide additional experimental support for K⁺ binding to Glt_{ph} (p. 8, line 3), it cannot distinguish between ionic conditions that promote translocation and those that prevent translocation on its own. Although this distinction can be easily done by radioactive uptake measurements (Ryan *et al. J Biol Chem* 284, 17540-17548 (2009)), we decided to remove the A233W quenching data from the main figures and moved them to Fig. S6.

We now interpret the results only as demonstration of K⁺ association to Glt_{ph} rather than as experimental indication for enhanced translocation and changed the subtitle of the next manuscript section from ‘*KI binding closes the extracellular gate and facilitates re-translocation*’ to ‘*KI binding closes the extracellular gate.*’

The authors appear to have overlooked a crucial consequence of their model. If GltPh in some conditions may counter transport K (even if it is not essential , and may happen only in a fraction of transport cycles), this will affect the reversal potentials. The Mindell group has recently published a method to do such measurements with prokaryotic transporters. The authors should do these experiments at different internal K concentrations. If the reversal potentials are dependent on the internal K concentration (e.g. between 1 and 300 mM, concentrations still sufficient for clamping of the voltage using valinomycin), it would be a strong argument in favor of the K binding hypothesis. If there is no different, K binding would not be relevant under physiological conditions.

We indeed considered using this elegant approach for our question. However, since there are no ionophores selective for other ions than K⁺, it is not applicable for studying K⁺ coupling in Glt_{ph}. When the liposomal membrane is clamped to the K⁺ reversal potential, the electrochemical gradient for K⁺ is zero during the experiment. Thus, K⁺-dependent and K⁺-independent transporters will experience the same driving force.

This at first glance surprising feature can be also demonstrated by calculating reversal potentials for K⁺-coupled and K⁺-independent Glt_{ph}.

If Glt_{ph} were additionally coupled to K⁺, the electrochemical gradient would be zero, when

$$\frac{[Glu]_o}{[Glu]_i} = \frac{[Na^+]_i^3 [K^+]_o}{[Na^+]_o^3 [K^+]_i} e^{\frac{FVr}{RT}}$$

$$e^{-\frac{FVr}{RT}} = \frac{[Glu]_i [Na^+]_i^3 [K^+]_o}{[Glu]_o [Na^+]_o^3 [K^+]_i}$$

$$Vr = -\frac{RT}{F} \ln \frac{[Glu]_i [Na^+]_i^3 [K^+]_o}{[Glu]_o [Na^+]_o^3 [K^+]_i}$$

In the suggested experimental setting, the membrane potential V_r is applied as K^+ diffusion potential and thus measures

$$V_r = \frac{RT}{F} \ln \frac{[K^+]_o}{[K^+]_i}$$

$$V_r = -\frac{RT}{F} \ln \frac{[Glu]_i [Na^+]_i^3 [K^+]_o}{[Glu]_o [Na^+]_o^3 [K^+]_i} = -\frac{RT}{F} \ln \frac{[Glu]_i [Na^+]_i^3}{[Glu]_o [Na^+]_o^3} - \frac{RT}{F} \ln \frac{[K^+]_o}{[K^+]_i}$$

$$= -\frac{RT}{F} \ln \frac{[Glu]_i [Na^+]_i^3}{[Glu]_o [Na^+]_o^3} - V_r$$

so that

$$V_r = -\frac{RT}{2F} \ln \frac{[Glu]_i [Na^+]_i^3}{[Glu]_o [Na^+]_o^3}$$

For the K^+ -independent Glt_{Ph} , the same result is obtained, since the number of moved charges per transport cycle increases from one to two

$$V_r = -\frac{RT}{2F} \ln \frac{[Glu]_i [Na^+]_i^3}{[Glu]_o [Na^+]_o^3}$$

Thus, the suggested experimental approach does not permit distinction between K^+ -dependent or K^+ -independent Glt_{Ph} .

Since Glt_{Ph} can translocate with or without bound K^+ , every assay quantifying the coupling stoichiometry will provide results intermediate to predictions for K^+ coupled or uncoupled transport. The exact numbers will depend on the relative transport rates for both conditions. These rates are currently not known, and our simulations do not provide any insights into K^+ -bound translocation.. Determination of the apparent K^+ coupling of Glt_{Ph} thus not allow experimental validation of simulation results.

Moreover, this information is not required for any conclusion of our manuscript. We demonstrate that Glt_{Ph} can bind K^+ (via simulations and experiments) and that K^+ promotes closure of HP2 (via simulations). These data suggest an allosteric coupling mechanism for K^+ in EAAT glutamate transport that we validate in experiments and simulations.

We did not intend to make any conclusion about the effect of K^+ on Glt_{Ph} function. K^+ -bound re-translocation might leave transport rates unaffected, enhance transport or even slow transport down. None of these possible outcome will have any effect on our conclusion about allosteric coupling in mammalian EAATs. We also did not make any conclusion about the physiological role of K^+ coupling in Glt_{Ph} . To prevent confusion of our future readers we edited the section of p. 14, line 21 accordingly.

On page 15 the authors describe conformational changed in the NMDGT motif upon K binding. There are structures of homologues (Jensen 2013 NSMB, Guskov 2016 Nature Comm) in the apo state , yet in the presence of K. Are the changes in conformation of the NMDGT motif

observed in these structures? Do the authors think these apo structure are infact K-bound. Discussion needed.

Thank you for this excellent comment! We now analyzed all existing crystal structures and found that the simulated K⁺-induced conformational change in the NMDGT motif in TM7 is similar to the conformational difference between the Glt_{ph} structure obtained in K⁺-,Na⁺-, and aspartate-free conditions (PDB ID: 4OYE) and the structures of Glt_{TK} purified in K⁺-containing and aspartate-free solution (PDB IDs: 4KY0, 5DWY; Fig. S8E). We describe and discuss this comparison on p.12, line 11.

Page 4: "Apo states were modelled by removing bound Na or Asp molecules from the structures". I do not understand why the authors have not used the published apo structures (Verdon, 2014, Jensen, 2013, Guskov, 2016

To maximize consistency of our simulation conditions, we decided to use Glt_{ph} as a model system for a K⁺-uncoupled Glt_X transporter, since crystallographic information on Glt_{ph} is available for both inward- and outward-facing states, in constrast to Glt_{TK}, which is only resolved in the outward-facing state (Jensen *et al.* (*Nat Struct Mol Biol* 20, 1224 - 1226 (2013)) and Guskov *et al.* (*Nat Comm* 7, 13420 (2016))). Furthermore, we choose Glt_{ph} over Glt_{TK} as more functional data are available on the former.

As a starting structure for Glt_{ph} simulations, we used the fully-bound Glt_{ph} structures (PDB IDs: 2NWX and 3KBC) as these have higher resolution and are not affected by the R397A mutation at the substrate-binding pocket, in contrast to Glt_{ph} structures obtained in Na⁺-only and apo states (Verdon *et al.* (*eLife* 3, e02283 (2014))). Before starting any production simulations, all simulations system have been extensively equilibrated during multiple hundreds of nanoseconds of MD to ensure reproducible structural relaxation in the respective occupancy states.

We now explain this on p.16, line 15.

We would like to thank the referees for their helpful criticisms, and we are grateful for the opportunity to submit a revised version of our manuscript. We look forward to your decision regarding its publication in *EMBO Journal*. Please contact me if we can provide additional information or if there are other concerns.

2nd Editorial Decision

12th Jun 2019

Thank you for submitting your revised manuscript for consideration by The EMBO Journal. Please accept my apologies for the delay in processing your revised manuscript due to protracted referee input. Your revised study was sent back to the three referees for re-evaluation, and we have received comments from two of them, which I enclose below. Please note that while referee #1 was unfortunately not able to re-assess the study at this time, we have considered your response to his/her raised critique and found these to be reasonably answered. As you will see the other referees find that their concerns have been sufficiently addressed and they are now broadly in favour of publication.

Thus, we are pleased to inform you that your manuscript has been accepted in principle for publication in The EMBO Journal, pending some minor issues related to the remaining discussion points of referee #3, as well as a number of formatting and data representation points listed below, which need to be adjusted at re-submission.

REFeree REPORTS:

Referee #2:

I am happy with the changes made to the manuscript

Referee #3:

There are two points remaining:

1. HP2 opening in IFC. In my view the authors are cherry picking published data and steering the interpreting of data published by others to support their own claim. To be short: there is no experimental evidence that HP2 opens on the cytoplasmic side. The only structural evidence for the inward gating element suggests that it could be HP1 (and not HP2) (Verdon elife). The authors selectively ignore this notion. In contract, they use the observation that the thermodynamics of TB(O)A binding is similar in IFC and OFC. In my view this is not prove for HP2 movement. The determinants for binding TB(O)A are poorly defined in the crystal structures, and there may not be much contribution to the binding energy neither from HP2 nor HP1, so either could be gating element. Finally, the statement "inhibitor binding to the IFC likely requires lateral motion of the transport domain to permit HP2 opening (Akyuz et al., 2015)" is difficult to place. I cannot find this statement in the Akyuz paper. I can only find that Akyuz (2015) and Garaeva (2018) indicate that HP2 might open in IFC, but they leave open the possibility that it could just as well be HP1. The authors must more carefully cite what has been discussed by others, and should make it much clearer that evidence for HP2 movement comes only from MD simulations.

2. The discussion on the A233W mutant on page 10 of the rebuttal is interesting, and should be moved to the main text. This is the type of observation that must be published, and not be hidden.

2nd Revision - authors' response

30th Jul 2019

The authors performed the requested editorial changes.

Corresponding Author Name: Christoph Fahlke

Manuscript Number: EMBOJ-2019-101468